# A highly potent ruthenium(II)-sonosensitizer and sonocatalyst for in vivo sonotherapy

Chao Liang[1,2,3], Jiaen Xie[1], Shuangling Luo[1], Can Huang[4], Qianling Zhang[1], Huaiyi Huang[4] & Pingyu Zhang [1✉]

As a basic structure of most polypyridinal metal complexes, $[Ru(bpy)_3]^{2+}$, has the advantages of simple structure, facile synthesis and high yield, which has great potential for scientific research and application. However, sonodynamic therapy (SDT) performance of $[Ru(bpy)_3]^{2+}$ has not been investigated so far. SDT can overcome the tissue-penetration and phototoxicity problems compared to photodynamic therapy. Here, we report that $[Ru(bpy)_3]^{2+}$ is a highly potent sonosensitizer and sonocatalyst for sonotherapy in vitro and in vivo. $[Ru(bpy)_3]^{2+}$ can produce singlet oxygen ($^1O_2$) and sono-oxidize endogenous 1,4-dihydronicotinamide adenine dinucleotide (NADH) under ultrasound (US) stimulation in cancer cells. Furthermore, $[Ru(bpy)_3]^{2+}$ enables effective destruction of mice tumors, and the therapeutic effect can reach deep tissues over 10 cm under US irradiation. This work paves a way for polypyridinal metal complexes to be applied to the noninvasive precise sonotherapy of cancer.

[1] College of Chemistry and Environmental Engineering, Shenzhen University, Shenzhen, P. R. China. [2] Key Laboratory of Optoelectronic Devices and Systems of Ministry of Education and Guangdong Province, College of Physics and Optoelectronic Engineering, Shenzhen University, Shenzhen, P. R. China. [3] College of Chemistry and Environmental Engineering, Hanshan Normal University, Chaozhou, P. R. China. [4] School of Pharmaceutical Sciences (Shenzhen), Sun Yat-sen University, Guangzhou, P. R. China. ✉email: p.zhang6@szu.edu.cn

The successful anticancer application of cis-platinum has greatly promoted the development of transition metal complexes as anticancer drugs[1–4]. However, due to its drug resistance and high toxicity, novel metallic drugs are urgently needed. Recently, metal-based photosensitizers were widely applied for photodynamic therapy (PDT), owing to their high photostability and $^1O_2$ generation[5–13]. However, one of the bottlenecks of PDT is the low tissue penetration depth of light. To enhance PDT efficiency, near-infrared two-photon and three-photon excitation have been employed, to increase the penetration depth of excitation light[14,15]. Nevertheless, the multi-photon light source is rather costly and troublesome in moving, thus it is not suitable for in vivo or clinical experiments[16,17]. In addition, multi-photon photo-dynamic therapy needs high power irradiation intensity with high phototoxicity. Therefore, it is urgent to discover sensitization mode for metal complexes to treat tumors deep in the tissue.

Sonodynamic therapy (SDT) is a non-invasive therapeutic strategy, which is triggered by the high tissue penetration ultra-sound (up to 10 cm)[18–21]. Similar to PDT, sonosensitizers can be activated to generate cytotoxic reactive oxygen species (ROS), such as •OH and $^1O_2$[22,23]. The sonosensitizers play a significant role in SDT. From early organic molecules such as various types of porphyrin derivatives to current inorganic nanoparticles have been developed to sonosensitizers[24], including ones assembled from organic molecules and inorganic nanoparticles such as MnTTP-HSA NPs[25], Si-based nanomaterials[26], and blackphosphorus-based sonosensitizers[27]. However, organic sonosensitizers often exhibit limited stability under US irradiation[28–30], and some of complicated nanoparticles show low ROS thus quantum yield decreasing SDT efficiency[31]. Taken together, an effective sonosensitizer agent is an important SDT precondition.

In addition to generation of ROS, oxidation of cellular bioactive small molecular such as NADH drew a lot attention recently. NADH is an important coenzyme, which participates in over 400 intracellular redox reactions[32–34]. NADH is produced by glycolysis and citric acid cycle in cellular respiration, and is considered as the carrier of biological hydrogen and electron donor in living cells. NADH is also related to the preservation of the redox balance within cells as well as prevents ROS related cell damages. For example, NADH can neutralize reactive oxygen species during PDT lead to poor efficacy[35,36]. Previously, we have implied that the metal-based photosensitizer could photo-oxidized NADH and destroy the redox balance[32]. To the best of our knowledge, there is no report concerning the relationship of SDT and NADH consumption.

$[Ru(bpy)_3]^{2+}$ is a basic and classical structure of polypyridinal metal complexes. The energy interval between LUMO and HOMO of $[Ru(bpy)_3]^{2+}$ is only 0.1239[37], suggesting $[Ru(bpy)_3]^{2+}$ is easy to be excited into the highly oxidative excited-state species. Sharipov group have studied that the energy of part of sonoluminescence (300–452 nm) of water is high enough to excite $[Ru(bpy)_3]^{2+}$. Moreover, radical products of sonolysis of water can also excite $[Ru(bpy)_3]^{2+}$. The sonochemiluminescence spectra of $[Ru(bpy)_3]^{2+}$ was recorded by US irradiation of argon saturated aqueous solutions of $[Ru(bpy)_3]^{2+}$[38–40].

In this work, we report that $[Ru(bpy)_3]^{2+}$ can be excited under US irradiation. In the presence of oxygen, US excited $[Ru(bpy)_3]^{2+}$ can transfer energy to oxygen to produce $^1O_2$. $[Ru(bpy)_3]^{2+}$ is almost non-cytotoxic without US stimulation. In contrast, $[Ru(bpy)_3]^{2+}$ exhibits excellent US-triggered $^1O_2$ production and sono-cytotoxicity ($IC_{50} = 2.91\ \mu M$). In addition, under US stimulation, $[Ru(bpy)_3]^{2+}$ oxidizes NADH into $NAD^+$, which is favorable for disruption of the redox balance in tumor cells. At the in vivo level, $[Ru(bpy)_3]^{2+}$ is adopted to generate the deep-tissue ROS in simulative tissue and in vivo, and its highly effective SDT treatment of tumors is achieved. This work opens a way for the application of various metal complexes in the noninvasive sono-therapy of cancer.

## Results

**Sonosensitizer performance.** $[Ru(bpy)_3]^{2+}$ has great advantages for research application such as simple structure, easy synthesis, and high yield. The MS, UV-visible absorption, $^1H$ NMR, $^{13}C$ NMR and emission spectra of $[Ru(bpy)_3]^{2+}$ were characterized in the supporting information (Supplementary Fig. 1). To investigate whether $[Ru(bpy)_3]^{2+}$ could use for SDT, we firstly employed electron spin resonance (ESR) to detect the ROS generation. The spin trap 2,2,6,6-tetramethylpiperide (TEMP) as the $^1O_2$ trapping agent. As shown in Fig. 1a, we can observe there 1:1:1 intensity signals appeared between 3480 and 3530 GM in TEMP and $[Ru(bpy)_3]^{2+}$ mixing solution under US irradiation. In contrast, the ESR spectra of water solvents alone by US irradiation were studied as Supplementary Fig. 2. We did not find obvious $^1O_2$ and •OH generation in water solution under US irradiation. Thus, $[Ru(bpy)_3]^{2+}$ excited by radical products of sonolysis of water was not the dominant mechanism of $[Ru(bpy)_3]^{2+}$ excited by US.

We further measured the quantum yield of $^1O_2$ by the oxidation of 9, 10-diphenanthraquinone (DPA)[41,42]. In the presence of $^1O_2$,

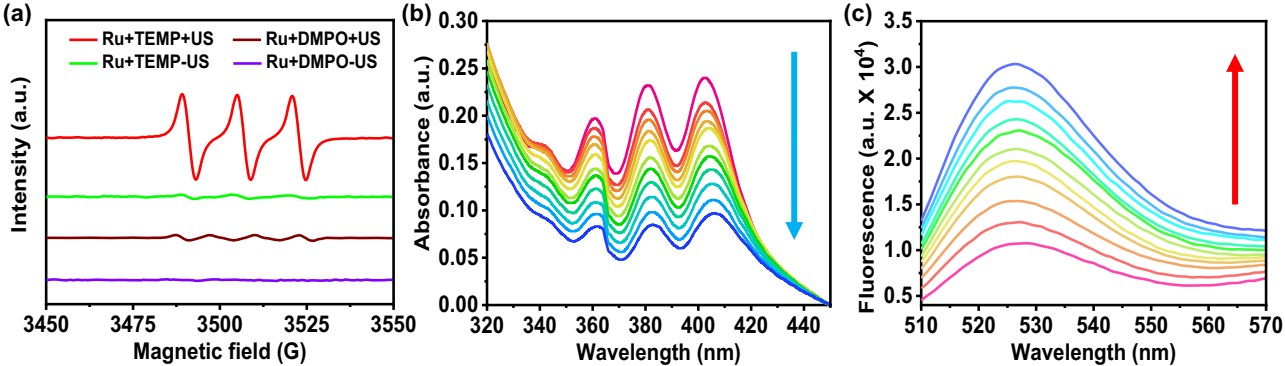

**Fig. 1 Sonosensitizer performance of $[Ru(bpy)_3]^{2+}$. a** ESR spectra demonstrating $^1O_2$ generation of $[Ru(bpy)_3]^{2+}$ under US irradiation (0.3 W cm$^{-2}$, 3 MHz, 1 h). The TEMP and DMPO were used as $^1O_2$ and •OH trapping agents, respectively. **b** Time-dependent oxidation of DPA indicating $^1O_2$ generation by $[Ru(bpy)_3]^{2+}$ under US irradiation (0.3 W cm$^{-2}$, 3 MHz). **c** Time-dependent $^1O_2$ generation of $[Ru(bpy)_3]^{2+}$ detected by fluorescence intensity of SOSG under US irradiation (0.3 W cm$^{-2}$, 3 MHz). The colored lines represent spectra recorded every 10 min for 100 min in (**a**) and (**c**). Ru: $[Ru(bpy)_3]^{2+}$. TEMP: 2,2,6,6-tetramethylpiperide; DMPO: 5,5-dimethyl-1-pyrroline-N-oxide.

DPA would be oxidized to 9,10-diphenanthraquinone dioxide (DPO$_2$), which has no obvious absorption in visible light band. As shown in Fig. 1b, with the increasing of US irradiation time, the characteristic absorption of DPA decreased gradually. The absorption peak of DPA around 378 nm was collected to measure the rate constant for DPA oxidation, and we find DPA oxidation versus time revealed linear relationship with the calculated rate constant was 0.00142 min$^{-1}$ (Supplementary Fig. 3a). However, the DPA oxidation by [Ru(bpy)$_3$]$^{2+}$ without US irradiation or US alone was very slightly (Supplementary Fig. 4). In addition, singlet oxygen sensor green (SOSG) was used to track the capture of $^1O_2$, along with its unique maximum fluorescence intensity at 525 nm. As shown in Fig. 1c and Supplementary Fig. 3b, the SOSG fluorescence intensity strengthened by degrees when [Ru(bpy)$_3$]$^{2+}$ and US irradiation were applied.

In the other hand, for •OH trapping agent, 5,5-dimethyl-1-pyrroline-N-oxide (DMPO) was used instead of TEMP at the same conditions. The result showed no obvious •OH signal in [Ru(bpy)$_3$]$^{2+}$ sample under US irradiation (Fig. 1a). Moreover, methylene blue (MB) was used to track the capture of •OH[43]. From Supplementary Fig. 5, we could not observe any change of absorption curve of MB, which verified [Ru(bpy)$_3$]$^{2+}$ with US irradiation can hardly produce •OH. In addition, we investigated the sono-stability of [Ru(bpy)$_3$]$^{2+}$ after 1 h US irradiation by detecting its absorption spectra and emission spectra (Supplementary Fig. 6). The results showed that [Ru(bpy)$_3$]$^{2+}$ exhibited high sono-stability under US irradiation, suggesting that [Ru(bpy)$_3$]$^{2+}$ has the potential to be an excellent sono-sensitizer.

**Sonocatalytic oxidation of NADH.** NADH is an important coenzyme, which participates in over 400 intracellular redox reactions. The selective induction of NADH ruin the redox balance and exterminate cancer cells[44]. Therefore, we quantified the sonocatalytic oxidation of NADH by [Ru(bpy)$_3$]$^{2+}$ under US irradiation. With increasing time of US irradiation, the NADH (150 μM) characteristic absorption peak around 339 nm decreased obviously in the presence of 10 μM [Ru(bpy)$_3$]$^{2+}$ (Fig. 2a and Supplementary Fig. 7). The NADH oxidation turnover frequency (TOF) was 3.62 h$^{-1}$ counted from the disparity in NADH consistence following US irradiation. The characteristic absorption peak of NADH around 339 nm was collected to measure the rate constant for NADH oxidation, and we found NADH depletion versus time showed a first-order kinetics relationship, and the calculated rate constant was 0.0385 min$^{-1}$. Importantly, by adding NaN$_3$ as a $^1O_2$ scavenger, we found that the NADH depletion rate by [Ru(bpy)$_3$]$^{2+}$ based SDT was not affected (Supplementary Fig. 8), indicating that [Ru(bpy)$_3$]$^{2+}$ was a sonocatalyst.

To further confirm the NADH sono-oxidation, ESR was employed to trap radical intermediates during US irradiation. 5-(2,2-dimethyl-1,3-propoxycyclo-phosphoryl)−5-methyl-1-pyrroline-N-oxide (CYPMPO) was selected as carbon-centred free radical scavenger to detect NAD•. As shown in Fig. 2b, the signal of CYPMPO-NAD was detected by ESR in water solution contain NADH and [Ru(bpy)$_3$]$^{2+}$ under US irradiation. The result proposed NADH was converted to NADH$^{+•}$ through single electron transfer mechanism by shifting an electron to [Ru(bpy)$_3$]$^{2+}$, which was excited by US. Then NADH$^{+•}$ easily deprotonated and turn into NAD• radical, which later change into NAD$^+$ by intramolecular migration[32].

$^1$H NMR spectroscopy was further used to monitor the transformation between NADH and NAD$^+$. After being irradiated with US, NADH was transformed into its oxidized form NAD$^+$ in the presence of [Ru(bpy)$_3$]$^{2+}$ (Fig. 2c). Some new

peaks of NAD$^+$ at 8.31, 8.55, 8.99, 9.36, and 9.58 ppm were observed. In contrast, no new peaks of NAD$^+$ in the NADH alone, Ru alone, Ru + NADH, and NADH + US control groups were found. On the other hand, we further investigated NADH depletion in 4T1 cells (Supplementary Fig. 9). Under US irradiation, the intracellular NADH concentration reduced after incubation with [Ru(bpy)$_3$]$^{2+}$, while only US irradiation or only [Ru(bpy)$_3$]$^{2+}$ incubation, the NADH levels were unaffected. The results confirmed that [Ru(bpy)$_3$]$^{2+}$ upon US irradiation could induce NADH oxidation. As all the above results, [Ru(bpy)$_3$]$^{2+}$ can generate $^1O_2$ and induce sonocatalytic oxidation of NADH under US irradiation, and its probable mechanism is shown in Fig. 2d.

**In Vitro SDT of [Ru(bpy)$_3$]$^{2+}$.** To evaluate in vitro sonotherapy efficiency of [Ru(bpy)$_3$]$^{2+}$, the methyl thiazolyl tetrazolium (MTT) assay was used to measure cytotoxicity of 4T1 murine breast cancer cells. Without US irradiation, [Ru(bpy)$_3$]$^{2+}$ exhibited no cytotoxicity in high concentrations (IC$_{50}$ > 160 μM) for 48 h incubation (Supplementary Fig. 10a). For SDT evaluation, 4T1 cells were incubated with various concentrations of [Ru(bpy)$_3$]$^{2+}$ (0–20 μM) for 4 h, followed by US irradiation for different time durations (0–25 min). The US power of 0.3 W cm$^{-2}$ was selected due to the temperature increase of the solution under higher US power (>0.3 W cm$^{-2}$). For example, US irradiation with the power of 0.4 W cm$^{-2}$ showed obvious heating effect on aqueous solution, and the temperature was high enough to kill 4T1 tumor cells directly (Supplementary Fig. 11a–c). In addition, the 4T1 cells killing efficiency decrease under lower power (0.1 W cm$^{-2}$ and 0.2 W cm$^{-2}$) of US irradiation (Supplementary Fig. 10b). And we chose 4 h incubation time of [Ru(bpy)$_3$]$^{2+}$ because it was well uptake by cells after 4 h, as shown in Supplementary Fig. 12. With US irradiation, the 4T1 cells viabilities continuously decreased with increasing concentration of [Ru(bpy)$_3$]$^{2+}$ (Fig. 3a). The IC$_{50}$ value was calculated as about 2.91 μM. The same phenomenon was also observed in another cytotoxicity experiment designed as the same [Ru(bpy)$_3$]$^{2+}$ concentration (10 μM) and different US irradiation time (Fig. 3b). Moreover, to intuitive display the sono-cytotoxicity of [Ru(bpy)$_3$]$^{2+}$ on 4T1 cells, the treated 4T1 cells were co-stained with calcein AM (AM) and propidium iodide (PI) (Fig. 3c). As expected, only [Ru(bpy)$_3$]$^{2+}$ incubation or only US irradiation showed strong AM signal and weak PI signal, which meant little damage to 4T1 tumor cells. But the cells in [Ru(bpy)$_3$]$^{2+}$ and US irradiation group showed faint green fluorescence from AM and intense red fluorescence from PI. These results proved that [Ru(bpy)$_3$]$^{2+}$ exhibited high sono-cytotoxicity toward tumor cells.

**ROS generation in cells.** To investigate cellular ROS of [Ru(bpy)$_3$]$^{2+}$ for sonotherapy, 2,7-dichlorofluorescein diacetate (DCFH-DA) and SOSG staining assays were used to confirm the intracellular ROS levels, respectively (Fig. 3d and Supplementary Fig. 13). The control, [Ru(bpy)$_3$]$^{2+}$ alone and US irradiation alone groups exhibited weak SOSG and DCFH-DA signal. In contrast, [Ru(bpy)$_3$]$^{2+}$ and US treated group showed strong green fluorescence from SOSG or DCFH-DA. The SOSG signal of [Ru(bpy)$_3$]$^{2+}$ and US treated group was decreased when NaN$_3$ (a $^1O_2$ scavenger) is present. The cytotoxicity of [Ru(bpy)$_3$]$^{2+}$ for sonotherapy on 4T1 cells was partly inhibited by adding NaN$_3$ (a $^1O_2$ scavenger) (Fig. 3c). These results suggested a large amount of intracellular $^1O_2$ was produced and then kill cancer cells. To investigate the kinds of intracellular ROS of [Ru(bpy)$_3$]$^{2+}$ for sonotherapy, dihydroethidium (DHE) and 3′-hydroxy-6′-(4-hydroxyphenoxy) spiro[2-benzofuran-3,9′-xanthene]-1-one (HPF) staining assays were used to capture superoxide anion (O$_2^{-•}$) and hydroxyl radical (•OH),

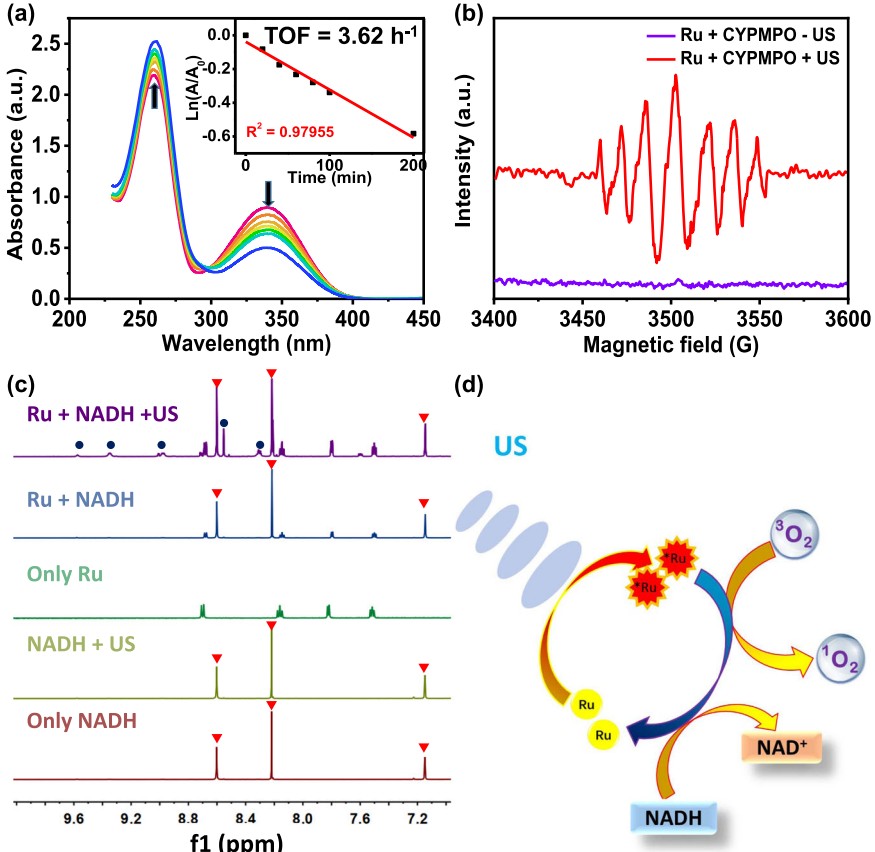

**Fig. 2 Sonocatalytic oxidation of NADH by [Ru(bpy)₃]²⁺ under US irradiation. a** The oxidation of NADH (150 μM) by [Ru(bpy)₃]²⁺ (10 μM) under US irradiation in PBS solution, as monitored by ultraviolet–visible spectroscopy. The direction of change in absorbance with time is indicated by the arrows. US irradiation time: 0, 20, 40, 60, 80, 100, 200 min. Insert: Plots of lnA/A₀ at 339 nm against time. **b** ESR spectrum of NAD• radicals trapped by CYPMPO demonstrating NADH oxidized by [Ru(bpy)₃]²⁺ under US irradiation. CYPMPO (1 mg) was used for NAD• radicals in 50 μL PBS solution containing [Ru(bpy)₃]²⁺ (5 mM) and NADH (10 mM) under US (0.3 W cm⁻², 3 MHz, 1 h) irradiation. **c** Sonocatalytic oxidation of NADH (1.1 mM) by [Ru(bpy)₃]²⁺ (0.1 mM) in D₂O/CD₃OD (1:1, v/v) with US irradiation (0.3 W cm⁻², 3 MHz, 20 min). Peaks labeled with red triangles represent NADH and peaks labeled with blue circles represent NAD⁺. **d** The proposed mechanism of ¹O₂ generation and NADH sonocatalytic oxidation by [Ru(bpy)₃]²⁺ under US irradiation. Ru: [Ru(bpy)₃]²⁺; TOF: turnover frequency; CYPMPO: 5-(2,2-dimethyl-1,3-propoxycyclo-phosphoryl)−5-methyl-1-pyrroline-N-oxide; NADH: 1,4-dihydronicotinamide adenine dinucleotide.

respectively (Supplementary Fig. 14). No obvious DHE signal or HPF signal could be found in the 4T1 cells treated with [Ru(bpy)₃]²⁺ and US irradiation. These results excluded the effects of SDT on O₂⁻• and •OH.

**ROS generation in deep-tissue.** Differ from PDT limited by the tissue penetration of light, sonotherapy is a promising new approach for deep-tissue tumor treatment due to excellent energy transfer efficiency of US. To investigate the sonotherapy efficiency in deep-tissue, 4T1 tumor-bearing mice were i.t. injected with SOSG and [Ru(bpy)₃]²⁺ mixing solution and irradiated by US on the other side of the mice, which was far from the tumor side. The ¹O₂ generation of tumor tissue was detect by an in vivo fluorescence imaging system (Fig. 4a). After US irradiation, the fluorescence signal of SOSG in the tumor tissue was obvious, suggesting that the [Ru(bpy)₃]²⁺ could generated ¹O₂ in deep tissue by US irradiation. Furthermore, a piece of pork (>10 cm) was selected to simulate human tissue for SDT-activatable depths research. SOSG and [Ru(bpy)₃]²⁺ mixing solution was injected at different distances (every 2 cm position) from the US probe into pork. As shown in Fig. 4b, after US irradiation, the ¹O₂ generation was detected even up to 10 cm away from the US probe.

To further confirm the generation of ROS in tumor tissue, 4T1 tumor-bearing mice were sacrificed 2 h post various treatments.

Their 4T1 tumor tissues were gathered for frozen sections and then stained by DCFH-DA and 4′,6-diamidino-2-phenylindole (DAPI), then these 4T1 tumor slices were photographed using a laser scanning confocal microscopy (LSCM) (Fig. 4c). Tumor tissue in [Ru(bpy)₃]²⁺ + US group demonstrated intense green fluorescence due to sufficient ROS generated by [Ru(bpy)₃]²⁺ based SDT. In contrast, the tumor sliced in the control, US and [Ru(bpy)₃]²⁺ alone groups exhibited very weak green fluorescence signal. All results certificate that [Ru(bpy)₃]²⁺ can generate ROS in deep-tissue in vivo.

**Sonodynamic therapy in Vivo.** Encouraged by the high ¹O₂ generation and sonocatalytic oxidation of NADH by [Ru(bpy)₃]²⁺, we further investigated antitumor efficacy in 4T1 tumor-bearing mice model. The mice were divided into four groups (5 mice per group): (1) Untreated; (2) [Ru(bpy)₃]²⁺ alone (i.t. injection 0.5 mg kg⁻¹); (3) US alone (0.3 W cm⁻², 3 MHz, 20 min); (4) [Ru(bpy)₃]²⁺ + US 0.1 (i.t. injection 0.5 mg kg⁻¹; 0.1 W cm⁻², 3 MHz, 20 min); (5) [Ru(bpy)₃]²⁺ + US 0.2 (i.t. injection 0.5 mg kg⁻¹; 0.2 W cm⁻², 3 MHz, 20 min); (6) [Ru(bpy)₃]²⁺ + US 0.3 (i.t. injection 0.5 mg kg⁻¹; 0.3 W cm⁻², 3 MHz, 20 min). After 4 h post i.t. injection of [Ru(bpy)₃]²⁺, the tumors were exposed by US irradiation. US at this power intensity was no thermal effect to kill tumor cells (Supplementary Fig. 11d, e). After that, the tumors were monitored

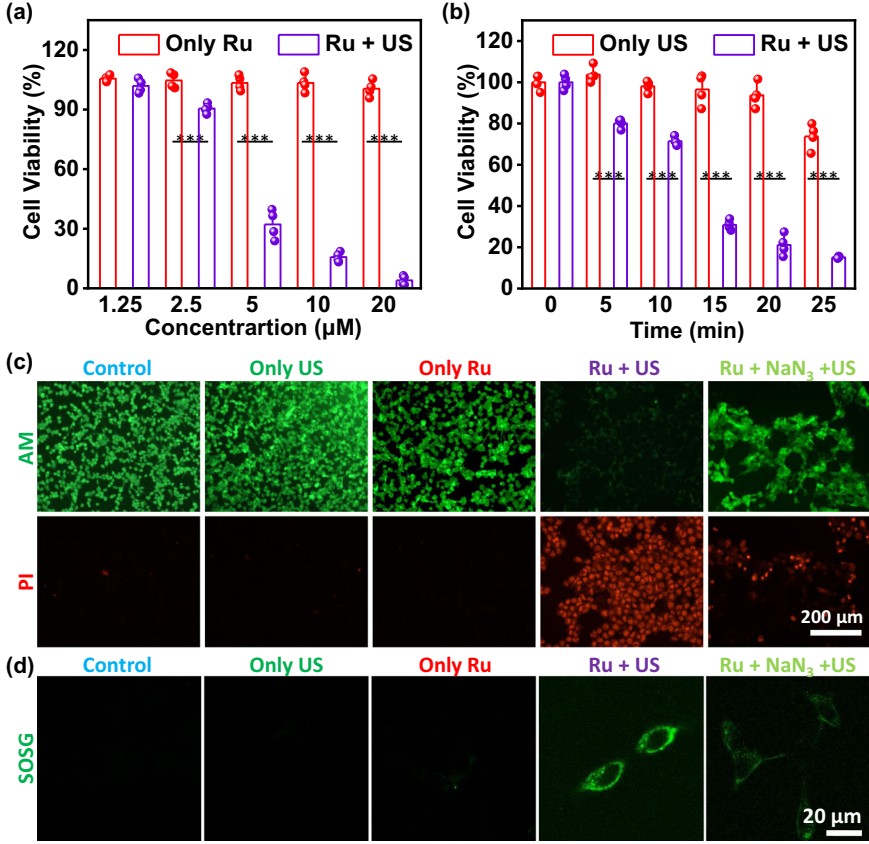

**Fig. 3 In vitro sonotherapy efficiency. a** The cell viabilities of 4T1 cells after incubation with different concentrations of $[Ru(bpy)_3]^{2+}$ in the presence or absence of US. **b** The cell viabilities of 4T1 cells treated with $[Ru(bpy)_3]^{2+}$ (10 μM) for varied US irradiation time. **c** Confocal images of 4T1 cells stained with calcein AM (green, live cells) and propidium iodide (red, dead cells) after different treatments. The experiment was repeated three times independently with similar results. **d** Confocal images of 4T1 cells stained with SOSG (green) after various treatments. Ru: $[Ru(bpy)_3]^{2+}$; US: 0.3 W cm$^{-2}$, 3 MHz, 20 min; AM: calcein AM; PI: propidium iodide. All cell viability data was performed as duplicates of quadruplicate ($n = 4$ biologically independent samples). Error bars represent S.D. from the mean. Statistical significance was calculated with two-tailed Student's $t$ test (**a**) and (**b**) (***$p < 0.001$, **$p < 0.01$, or *$p < 0.05$).

by digital caliper and their volumes were calculated by the formula: Volume = 0.5 * Length * Width$^2$ (Fig. 5a). The tumor growth was remarkably suppressed in $[Ru(bpy)_3]^{2+}$ + US 0.3 group, while tumors in the untreated group, $[Ru(bpy)_3]^{2+}$ alone group and US alone group showed obvious growth (Fig. 5b, c). In addition, 0.3 W cm$^{-2}$ US irradiation showed improved inhibitory effect on tumors growth than 0.1 W cm$^{-2}$ and 0.2 W cm$^{-2}$ US irradiation. At the end of experiment, the mice in different groups were sacrificed so that the tumors can be gathered to photograph and weigh (Fig. 5d, e). Among the four groups, the average tumor weight in $[Ru(bpy)_3]^{2+}$ + US 0.3 group was the least (Fig. 5e). To confirm the efficient SDT in the deep-tumor tissues, the 4T1 tumor was transplanted on the right side of mice, and the US probe was on the left side of mice during treatment. The US wave penetrated from left to right. The result was shown in Supplementary Fig. 15, confirming that the efficient SDT of $[Ru(bpy)_3]^{2+}$ can reach deep-tumor tissues.

Furthermore, the sonotherapy efficacy was evaluated by haematoxylin and eosin (H&E) staining assay and TdT-mediated dUTP nick-end labeling (TUNEL) assay (Fig. 6a). 24 h after various treatments, the mice were sacrificed to collect their tumors for histological analysis. Consistent with the above data, we observed severe tissue damage of the tumor tissue in $[Ru(bpy)_3]^{2+}$ and US treated group. In contrast, control, only $[Ru(bpy)_3]^{2+}$ or only US group showed no obviously tissue damage confirmed by both H&E and TUNEL staining slices. The tumor tissue in NaN$_3$ and $[Ru(bpy)_3]^{2+}$ added with US irradiation group showed decrease tissue damage compared to

$[Ru(bpy)_3]^{2+}$ and US treated group, but obviously tissue damage compared to the other three groups. The results suggested that SDT damage to tumor tissue was related to singlet oxygen generation (Fig. 6a). The biosafety of $[Ru(bpy)_3]^{2+}$ were evaluated by H&E stained slices of main organ collected from healthy mice i.v. injected with five times of the therapeutic dose (2.5 mg kg$^{-1}$) of $[Ru(bpy)_3]^{2+}$ (Supplementary Fig. 16). The results showed that no obvious tissue damage was found from these slices. We further studied that the LD$_{50}$ of $[Ru(bpy)_3]^{2+}$ was 3.89 mg kg$^{-1}$ in the acute toxicity experiment. These results mean that $[Ru(bpy)_3]^{2+}$ is safe in vivo.

**Anti-metastasis to lung in vivo**. Lung metastasis is usually discovered in advanced cancer, which result in a rapid death. 40 days following the mentioned four treatments, India-ink was tracheal injected to darken healthy alveoli and their 1 mm-thick horizontal sections of lung were collected to photograph. As shown in Fig. 6b, a large number of lung metastasis sites (the white tissue was circled) were observed in lungs collected from untreated, $[Ru(bpy)_3]^{2+}$ alone and US alone groups. In marked contrast, lung collected from $[Ru(bpy)_3]^{2+}$ + US group showed no metastasis site. Moreover, obvious tumor characteristic tissue (crowded cancer cells) could be found in the H&E stained lung slice collected from untreated, $[Ru(bpy)_3]^{2+}$ alone and US alone groups (Fig. 6c). These results indicated that $[Ru(bpy)_3]^{2+}$ based sonotherapy inhibited the progress of tumor lung metastasis.

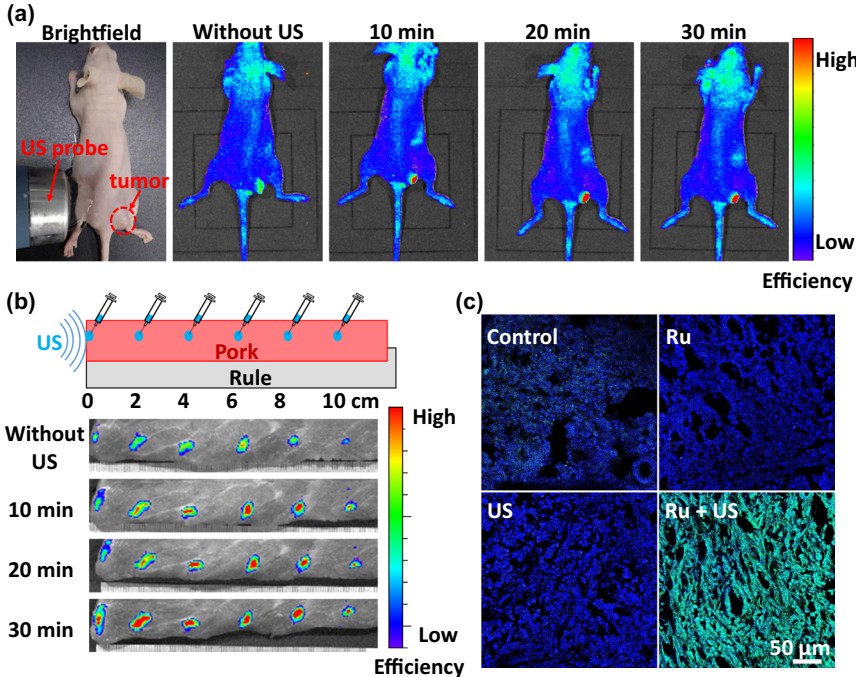

**Fig. 4 ROS generation in deep tissue. a** The fluorescence imaging of 4T1 tumor-bearing nude mice with i.t. injection of SOSG and [Ru(bpy)$_3$]$^{2+}$ for varied US irradiation durations. The tumor tissue position is pointed out by red circle. **b** The fluorescence imaging to investigate $^1O_2$ generation in the presence of [Ru(bpy)$_3$]$^{2+}$ and SOSG in deep pork tissue (>10 cm) under 30 min US irradiation. SOSG and [Ru(bpy)$_3$]$^{2+}$ mixing solution was injected at different distances (every 2 cm position) from the US probe into pork. **c** The fluorescence imaging of DAPI (blue) and DCFH-DA (green) co-stained tumor slices collected from mice after different treatments. Ru: [Ru(bpy)$_3$]$^{2+}$; US: 0.3 W cm$^{-2}$, 3 MHz.

## Discussion

In summary, this work performed a polypyridinal ruthenium(II) complex for sonodynamic therapy and amplified the use of metal complexes in anti-tumor therapy. Compared with PDT, SDT exhibited deep tissue penetration and unique US activation. In this study, [Ru(bpy)$_3$]$^{2+}$, a simple classical polypyridine metal complex but a highly potent sonosensitizer and sonocatalyst, shows efficient $^1O_2$ generation and NADH sonocatalytic oxidation, sharply differs from traditional sono-sensitizers including hematoporphyrin, photofrin, chlorin and phthalocyanine[45–49]. Under US irradiation, [Ru(bpy)$_3$]$^{2+}$ oxidize intracellular NADH, which could destroy redox balance in tumor cells, thus greatly raise SDT efficiency. At the in vivo level, [Ru(bpy)$_3$]$^{2+}$ based sonotherapy effectively inhibited tumor growth and metastasis. Importantly, simply, clear, substitution-inert structure and non-dark toxicity of [Ru(bpy)$_3$]$^{2+}$ reduce public worry about heavy metal toxicity. This work would provide an idea of noninvasive and widely applicable metal-based sonosensitizers or sonocatalyst for sonotherapy of tumor. It is a pity that the Ru drug can only use intratumoral injection at the moment. In the future, our work will focus on delivering [Ru(bpy)$_3$]$^{2+}$ to tumor tissue after systemic injection instead of local injection using [Ru(bpy)$_3$]$^{2+}$ loaded on liposome or protein to improve tumor-targeting in vivo.

## Methods

**Analysis of $^1O_2$ generation.** 10% DMSO and 90% H$_2$O solution with 5 μM [Ru(bpy)$_3$]$^{2+}$ and 2 μg mL$^{-1}$ DPA was measured using UV-vis spectrophotometer after different US (0.3 W cm$^{-2}$, 3 MHz) irradiation time. The absorbance changes of DPA at 378 nm were recorded to calculate the generation rate of $^1O_2$. The other method, 5 μM SOSG and 5 μM [Ru(bpy)$_3$]$^{2+}$ mixing solution was measured using fluorescence spectrometer after different time of US irradiation. The fluorescence changes of SOSG at 525 nm were recorded.

**Analysis of •OH generation.** 5 μM [Ru(bpy)$_3$]$^{2+}$ and 5 μg mL$^{-1}$ methylene blue (MB) mixing solution was measured using UV-vis spectrophotometer with

increasing US irradiation time. The absorbance changes of MB were recorded to analyze the generation of •OH.

**ESR measurements.** ESR measurements were carried out on a Bruker Model A300 ESR spectrometer equipped with a Bruker ER 4122 SHQ resonator, using 1.0 mm quartz tubes. The TEMP and DMPO were used to detect $^1O_2$ and •OH, respectively. 20 μL TEMP (40 mM) or DMPO (90 mM) was mixed with 80 μL [Ru(bpy)$_3$]$^{2+}$ (5 mM) and irradiated by US for 1 h. As a comparison, the [Ru(bpy)$_3$]$^{2+}$ mixed with TEMP or DMPO without US group were detected as well. Without [Ru(bpy)$_3$]$^{2+}$, TEMP and DMPO solution irradiated by US were detected as control groups. CYPMPO (1 mg) was used for detecting NAD• radicals in [Ru(bpy)$_3$]$^{2+}$ (5 mM) and NADH (10 mM) mixing solution under US irradiation.

### Oxidation of NADH under US irradiation

*UV-vis absorption method.* PBS solution including 10 μM [Ru(bpy)$_3$]$^{2+}$ and 150 μM NADH was measured using UV-vis spectrophotometer post different time of US irradiation. The absorbance changes of NADH at 339 nm were recorded to quantify the oxidation rate of NADH. Turnover Frequency (TOF) was calculated by dividing the difference in NADH concentration after 1 h US irradiation by the concentration of [Ru(bpy)$_3$]$^{2+}$.

*NMR method.* NADH (1.1 mM) was added to an NMR tube containing [Ru(bpy)$_3$]$^{2+}$ (100 μM) in CD$_3$OD and D$_2$O solution (1/1, v/v). Following 20 min of US irradiation, $^1$H NMR spectra of the resulting solutions were recorded at 310 K. As a comparison, [Ru(bpy)$_3$]$^{2+}$ + NADH without US irradiation, only [Ru(bpy)$_3$]$^{2+}$, NADH + US, and only NADH groups were recorded as well.

*Intracellular NADH detection.* 4T1 cells were seeded per well in 12-well plates for 24 h. After various treatments, the cellular NADH concentrations was determined using the NAD/NADH-Glo™ kit (Promega) by chemiluminescence using a micro-plate reader. Each group was determined as duplicates of quadruplicate.

**Sono-cytotoxicity.** 4T1 breast cancer cells were obtained from American Type Culture Collection (ATCC). 4T1 cells were incubated with various concentrations of [Ru(bpy)$_3$]$^{2+}$ (0–20 μM) for 4 h, followed by different powers of US irradiation (0–0.3 W cm$^{-2}$, 3 MHz) for different time durations (0–25 min). The cell viability of each group was detected using a MTT assay. For in vitro fluorescence imaging of live and dead cells, 4T1 cells were incubated with [Ru(bpy)$_3$]$^{2+}$ (10 μM) for 4 h, followed by US irradiation. After SDT, the 4T1 cells were co-stained with calcein AM (AM, live cell) and propidium iodide (PI, dead cell).

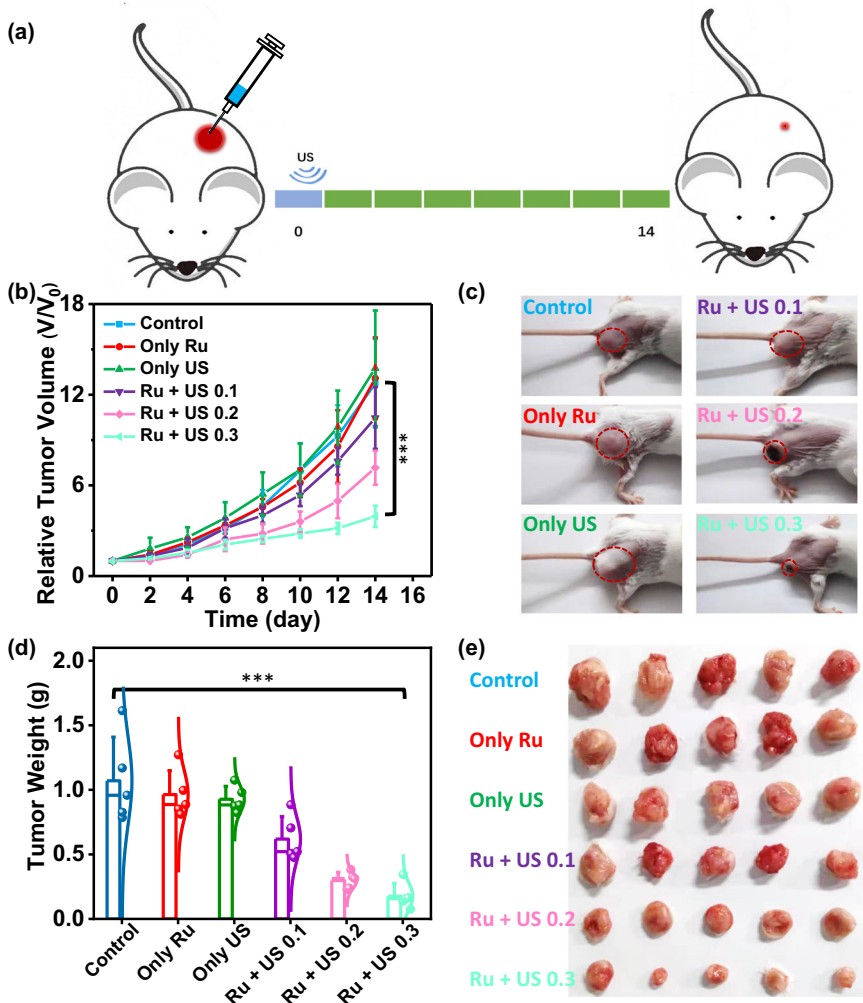

**Fig. 5 In vivo sonotherapy. a** Schematic of the in vivo sonotherapy procedure in 4T1 tumor-bearing mice. Mice was irradiated by US (0.1, 0.2, 0.3 W cm$^{-2}$, 3 MHz) for 20 min after 4 h i.t. injected with [Ru(bpy)$_3$]$^{2+}$ solution. Tumor sizes were monitored every two days for 14 days in total. **b** Tumor growth curves of mice after various treatments. Error bars were standard errors (±SD) based on five mice in each group. Statistical significance was calculated with two-tailed Student's $t$ test, $p = 0.000092$ (***$p < 0.001$, **$p < 0.01$, or *$p < 0.05$). **c** Representative images of mice at day 14 after various treatments. **d** Average tumor weights of mice at 14 day post various treatments as shown in (**e**). **e** Photos of tumors were collected from mice at 14 day after various treatments. Ru: [Ru(bpy)$_3$]$^{2+}$. Error bars were standard errors based on five mice in each group. Statistical significance was calculated with two-tailed Student's $t$ test, $p = 0.00023$ (***$p < 0.001$, **$p < 0.01$, or *$p < 0.05$).

**Intracellular ROS measurement**. The ROS in the cells was detected by SOSG or DCFH-DA. In detail, the 4T1 cells with various treatments were incubated with SOSG or DCFH-DA for 20 min, and then followed by US irradiation (0.3 W cm$^{-2}$, 3 MHz, 20 min). Finally, all the cell images were acquired using a Zeiss LSM 880 confocal microscopy.

**Tumor model**. Balb/c mice and nude mice were purchased from Liaoning Changsheng Biotechnology Co. Ltd. Mice were housed in individually ventilated cage (IVC) systems (ambient temperature: 23 ± 3 °C; relative humidity: 40–70%) and exposed to a 12-h light–dark cycle with free access to food and water. This work was conducted in according with Animal Care and Institutional Ethical Guidelines in China. And all animal experiments were carried out under the permission by the Ethic Committee of Shenzhen University (certificate number: SYXK 2014-0140). One million 4T1 cancer cells in 25 μL PBS were subcutaneously injected to the right back of each mice. About 7 days after injection, the mice with ~100 mm$^3$ tumor volume were selected for further experiments.

**$^1$O$_2$ generation in deep tissue**. To investigate the SDT depth in tissue, $^1$O$_2$ generation was detected after US irradiation in vivo. The SOSG and [Ru(bpy)$_3$]$^{2+}$ (10 μM) were i.t. injected in 4T1 tumor on the right of the mice. On the left side of the mice laid a US probe. After different US irradiation durations, SOSG signal was observed by an IVIS in vivo imaging system. On the other hand, a long piece of pork was selected as bionic human muscle tissue. The SOSG and [Ru(bpy)$_3$]$^{2+}$ (10 μM) were injected in disparate places of pork. The US probe was placed on the

left side of the pork. The rest of the steps were the same as the above-mentioned detection of mice. The intratumoral $^1$O$_2$ generation was also detected by fluorescence staining. 2 h after various treatment, 4T1 bearing mice were sacrificed to collect their tumors for frozen section. The tumor slices were stained with DIPA and DCFH-DA and imaged by a Leica confocal fluorescence microscope.

**In vivo sonotherapy experiments**. 4T1 tumor-bearing Balb/c mice were divided randomly into 6 groups ($n = 5$ per group) for distinct treatments: (1) Control, (2) Only Ru, (3) Only US, (4) Ru + US 0.1 W cm$^{-2}$ (5) Ru + US 0.2 W cm$^{-2}$ (6) Ru + US 0.3 W cm$^{-2}$; Ru: i.t. injection, 10 μg in 25 μL PBS per mice (500 μg Ru/1 kg mice weight); US: 3 MHz, 20 min. Tumor sizes were monitored every two days for 14 days. The tumor volumes were calculated by the formula: volume = 0.5 * length × width$^2$. 14 days after treatment, the mice were sacrificed and their tumors were gathered to photograph and weigh. For histology examination, at 24 h post treatment, tumor tissue was collected from different groups of mice. After fixing in 10% formalin, tumor tissue was paraffin embedded and sectioned for H&E and TUNEL staining.

For lung metastasis assay, the mice were injected with India ink into their lungs through the trachea after 40 days treatments. The mice were sacrificed to collect their lungs. The lungs were horizontally sliced and photographed after soaked in a Fekete's solution (5 mL glacial acetic acid, 10 mL formalin, 100 mL of 70% alcohol). India ink led to stained black in healthy alveolar tissue, by contrast, tumor metastasis sites appeared to be white. The collected lungs were then sectioned into slices with 8-micrometer thickness, and stained with hematoxylin and heosin.

In addition, in vivo sonotherapy experiments in deep tumor model were designed as Supplementary Fig. 15a. The 4T1 tumor was transplanted on the right

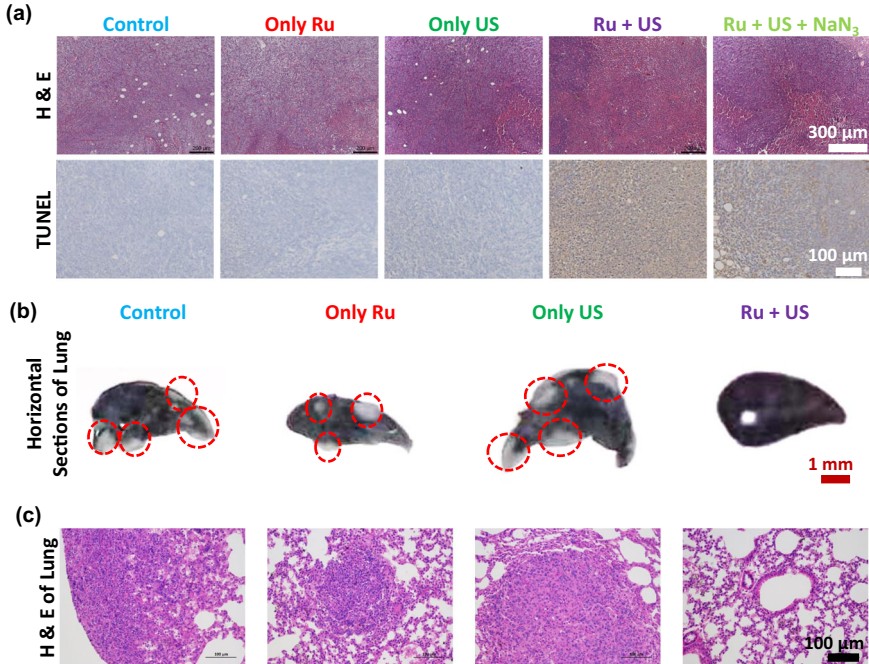

**Fig. 6 Tumor histology and lung metastasis examination. a** Microscopy photos of H&E and TUNEL stained tumor slices. Tumors tissue were collected from mice at 24 h post various treatments. The experiment was repeated three times independently with similar results. **b** Photos of horizontal sections of India ink stained lungs were collected from mice at 40 day post various treatments. Spontaneous pulmonary breast cancer metastasis sites are pointed out by red circles. **c** Microscopy images of H&E stained lung slices collected from different groups of mice at 40 day post various treatments. The experiment was repeated three times independently with similar results. US irradiation: 0.3 W cm$^{-2}$, 3 MHz; Ru: [Ru(bpy)$_3$]$^{2+}$.

side of mice, and the US probe was on the left side of mice during treatment. The US waves penetrated from left side of mice to right side during SDT, while other experimental details remain unchanged as before. For the biosafety evaluation, healthy mice were i.v. injected with 2.5 mg kg$^{-1}$ [Ru(bpy)$_3$]$^{2+}$. Mice were sacrificed at 1 or 7 day post injection to collect their main organ for H&E stained slices. In acute toxicity experiment of [Ru(bpy)$_3$]$^{2+}$, each i.v. injection dose was investigated in 6 mice.

**Reporting summary**. Further information on research design is available in the Nature Research Reporting Summary linked to this article.

## Data availability

The authors declare that all data needed to evaluate the conclusion of this work are presented in the paper, the Supplementary Information, or Source data file. The all data generated in this study have been deposited in the Figshare database under accession code DOI: 10.6084/m9.figshare.15028020. [https://figshare.com/articles/figure/Data_of_A_Highly_Potent_Ruthenium_II_Sonosensitizer_and_Sonocatalyst_for_in_Vivo_Sonotherapy_/15028020]. Other data related to this work are available from the corresponding authors upon reasonable request. Source data are provided with this paper.

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

## Acknowledgements

We appreciate the financial support of the National Natural Science Foundation of China (NSFC, 22077085, 21701113, 22007104, and 21671137), the Project of the Natural Science Foundation of Guangdong Province (2019A1515011958), the Science and Technology Foundation of Shenzhen (JCYJ20190808153209537), the Natural Science Foundation of SZU (2018036) and Peacock Talent Fund (827-000389), and the fellowship of China Postdoctoral Science Foundation (2019M663067). We appreciate the Instrumental Analysis Center of Shenzhen University.

## Author contributions

P.Z. and L.C. oversaw and designed all experiments; Q.Z. and H.H. synthesized the complex; J.X., S.L, and C.H. performed the sonodynamic experiments in the solution and in the cancer cells; L.C. and J.X. performed the animal experiments; P.Z., L.C., and J.X. wrote the paper. All authors reviewed and edited the paper.

## Competing interests

The authors declare no competing interests.
