## [Peer Review File · Nature Communications]

Reviewers' Comments:

Reviewer #1:

Remarks to the Author:

In this work, the authors for the first time employ $[\text{Ru}(\text{bpy})_3]^{2+}$ as potent sonosensitizer and sonocatalyst for in vitro and in vivo sonotherapy. $[\text{Ru}(\text{bpy})_3]^{2+}$ could be activated by US irradiation and then generate $^1\text{O}_2$ and sono-oxidize endogenous NADH. Lastly, this sonosensitizer shows effective therapeutic effect for mice tumor with 10 cm tissue penetration. Overall, this study is interesting and organized, and this work is important for the application of polypyridinal metal complexes for SDT. But some issues should be solved before publication:

1. Some data about the preparation and characterization of $[\text{Ru}(\text{bpy})_3]^{2+}$ should be involved in the manuscript.
2. Please investigate and discuss the reaction mechanism of US-activated $[\text{Ru}(\text{bpy})_3]^{2+}$ for $^1\text{O}_2$ generation.
3. There are many mistakes in the manuscript. Line 110-111, ESR spectra of DMPO-OH is not Fig. 1b, and this data is missed by the authors in the manuscript. Line 129-130, the NADH depletion curves is not Fig. 2b. Line 178-179, the sentence 'To investigate the sonotherapy.....with SOSG and $[\text{Ru}(\text{bpy})_3]^{2+}$ ' is repeating. The authors should carefully check the manuscript and correct these mistakes.
4. Please investigate and quantify the in vitro and in vivo NADH depletion by US-activated $[\text{Ru}(\text{bpy})_3]^{2+}$.
5. In Figure 4b, there is lack of a control group that SOSG + $[\text{Ru}(\text{bpy})_3]^{2+}$ without US irradiation. Please add.
6. How does the author conduct the US treatment in the 'in vivo sonotherapy' section? What's the distance of the US probe with the tumor during the US treatment? The authors claim that the therapeutic effect can reach deep tissues over 10 cm under US irradiation. Thus, please add a deep tumor model to confirm the efficient SDT performance of $[\text{Ru}(\text{bpy})_3]^{2+}$.
7. As for deep tumor model, systemic injection is more appropriate than local injection. Thus, can the authors achieve the SDT post i.v. injection of $[\text{Ru}(\text{bpy})_3]^{2+}$ with proper modification?
8. How about the biosafety and the acute toxicity of $[\text{Ru}(\text{bpy})_3]^{2+}$? Some in vivo toxicity data should be provided at least to support the feasibility of this new material for biomedical applications.

Reviewer #2:

Remarks to the Author:

The authors present an interesting paper about the sonosensitizing properties of Ruthenium(II) providing the results of its activation under US exposure on in vitro and in vivo models. The manuscript could be improved by considering the following revisions:

- The authors performed the in vitro sonodynamic treatment after a cell incubation of 4 h with Ruthenium(II) but a Ruthenium(II) cell uptake study was not provided. How did the authors select the Ruthenium(II) time of cell incubation for the sonodynamic experiments?
- The authors performed the in vivo sonodynamic treatment by an intratumoral administration of Ruthenium(II). Do the authors know the Ruthenium(II) biodistribution after e.v. or i.p. administration? Are there data about the systemic toxicity of Ruthenium(II)?
- The authors used high US frequency (3 MHz) and low US intensity (0.3 W cm^{-2}) for the insonation compared to the commonly used US parameters for in vivo sonodynamic treatment. Why did the authors select these US parameters?
- Which is the mechanism underpinning the Ruthenium(II) activation? Do the authors suppose the occurrence of acoustic cavitation or sonoluminescence? According to literature, the US frequency and intensity used by the authors are not so efficient in the generation of acoustic cavitation and sonoluminescence. The authors have to explain their hypothesis about the Ruthenium(II) activation by US in the discussion section, explaining the reason why they selected the US parameters used.
- About the Ruthenium(II) generation of $^1\text{O}_2$ and $^{\bullet}\text{OH}$ under US exposure, it would be very interesting to know what happens during US exposure alone (the ESR spectra of US alone are not shown in Figure 1 or in supplementary materials).
- What about the medium temperature during the ESR experiments that are performed by a long

time of US exposure?

- In Figure 3a and b the statistical significance has to be shown.
- There are some oversights throughout the manuscript.

Reviewer #3:

Remarks to the Author:

In this article, the author pioneered the use of $[\text{Ru}(\text{bpy})_3]^{2+}$ as a kind of sonosensitizer and tried to demonstrate that $[\text{Ru}(\text{bpy})_3]^{2+}$ -mediated sonodynamic therapy effectively killed the breast cancer cells in vitro and in vivo. The mechanism of cell death is mainly due to the generation of singlet oxygen promoted by sonodynamic therapy. The part of physics is well organized, but the evidence in biological section is obviously insufficient. This research just demonstrated the cell-killing effect of a therapy and its ability to produce singlet oxygen. Otherwise, the following questions puzzled me a lot. I think this study needs to be carefully planned again to promote the innovation.

1. According to the instructions of SOSG (Invitrogen, S36002), SOSG is cell-impermeable thus it's not proper to be used to test the singlet oxygen in cells or tissues. If your SOSG is available, please provide the brand and catalog number of the drug.
2. I wonder the basis for determining the duration and manner of $[\text{Ru}(\text{bpy})_3]^{2+}$ injection. Why you choose to expose tumors to US just at 10 min post injection of $[\text{Ru}(\text{bpy})_3]^{2+}$? No direct proof of $[\text{Ru}(\text{bpy})_3]^{2+}$ distribution within the tumor can be found in this passage. And why you choose the manner of intrathecal injection but not intravenous injection or intraperitoneal injection? If you have strong evidence, please mark the references.
3. According to Figure 3b, why is the duration of treatment determined at 20 min instead of 15 min? It's obvious that US for 20 min can cause a certain percentage of cell death.
4. The reactive oxygen species other than singlet oxygen and hydroxyl radicals were not excluded in this study.
5. There is no sufficient evidence to prove the connection between cell death and singlet oxygen generation except for one experiment on 4T1 cells using NaN_3 .
6. There are some mistakes in grammar and diction in the article as follows:
 - 1) In Page3 Line55 "intensit" should be changed to "intensity";
 - 2) In Page3 Line57-58 "Sonodynamic therapy (SDT) is a novel non-invasive strategy, which triggered by the high tissue penetration ultrasound (up to 10 cm)" should be changed to "Sonodynamic therapy (SDT) is a novel non-invasive strategy, which is triggered by the high tissue penetration ultrasound (up to 10 cm)";
 - 3) In Page3 Line64-66 "However, organic sonosensitizers often exhibit limited stability under US, and some of complicated nanoparticles show low ROS quantum yield decreasing SDT efficiency" should be changed to "However, organic sonosensitizers often exhibit limited stability under US, and some of complicated nanoparticles show low ROS quantum yield thus decreasing SDT efficiency";
 - 4) In Page4 Line70-72 "NADH is produced by glycolysis process and citric acid cycle in cellular respiration, and considers as the carrier of biological hydrogen and electron donor in living cells" should be changed to "NADH is produced by glycolysis process and citric acid cycle in cellular respiration, and is considered as the carrier of biological hydrogen and electron donor in living cells";
 - 5) In Page4 Line74, it is not proper to describe PDT as a kind of cell damage, PDT is a kind of therapy.
 - 6) In Page6 Line117-119 "The results showed that $[\text{Ru}(\text{bpy})_3]^{2+}$ exhibited a strong ability to produce $^1\text{O}_2$ under US and high sono-stability, suggesting that $[\text{Ru}(\text{bpy})_3]^{2+}$ has potential to be an excellent sono-sensitizer, and its probable mechanism was shown in Fig. 1d" should be changed to "The results showed that $[\text{Ru}(\text{bpy})_3]^{2+}$ exhibited strong ability to produce $^1\text{O}_2$ and high sono-stability under US, suggesting that $[\text{Ru}(\text{bpy})_3]^{2+}$ has the potential to be an excellent sono-sensitizer, and the probable mechanism was shown in Fig. 1d";
 - 7) In Page7 Line137-139 "The results suggested that NADH transforming into $\text{NADH}^+\bullet$ via single electron transfer mechanism by transferring an electron to US induced excited-state $[\text{Ru}(\text{bpy})_3]^{2+}$ " should be changed to "The results suggested that NADH was transformed into $\text{NADH}^+\bullet$ via single electron transfer mechanism by transferring an electron to $[\text{Ru}(\text{bpy})_3]^{2+}$ which is excited by US";

8) In Page7 Line143-146 "Then, irradiated with US (0.3 W cm⁻², 3.0 MHz) for 20 min, NADH was converted into its oxidized form NAD⁺ (with new peaks at 8.31, 8.55, 8.99, 9.36 and 9.58 ppm assignable to the hydrogen atoms of NAD⁺) was observed in the 1 H NMR spectrum" should be changed to "Then, after being irradiated with US (0.3 W cm⁻², 3.0 MHz) for 20 min, NADH was converted into its oxidized form NAD⁺ (with new peaks at 8.31, 8.55, 8.99, 9.36 and 9.58 ppm assignable to the hydrogen atoms of NAD⁺) which can be observed in the 1 H NMR spectrum";

9) In Page10 Line209-210, "Average tumor weight in [Ru(bpy)₃]²⁺ +US group was the least in all four groups and even a mouse was cure" should be changed to "Among the four groups, the average tumor weight in [Ru(bpy)₃]²⁺ +US group was the least and one of the five mice was completely cured".

We would like to thank all the reviewers for their insightful comments about our work. The following are our point-to-point responses to the concerns raised and revisions made accordingly.

Reply to Reviewer 1:

In this work, the authors for the first time employ [Ru(bpy)₃]²⁺ as potent sonosensitizer and sonocatalyst for in vitro and in vivo sonotherapy. [Ru(bpy)₃]²⁺ could be activated by US irradiation and then generate ¹O₂ and so-so-oxidize endogenous NADH. Lastly, this sonosensitizer shows effective therapeutic effect for mice tumor with 10 cm tissue penetration. Overall, this study is interesting and organized, and this work is important for the application of polypyridinal metal complexes for SDT. But some issues should be solved before publication:

Reply to the reviewer: We thank the reviewer for the enthusiastic comments. Our responses to these issues are attached as follows.

1). *Some data about the preparation and characterization of [Ru(bpy)₃]²⁺ should be involved in the manuscript.*

Reply to the reviewer and revisions made: We thank the reviewer for this suggestion. During the revision, we have added the following discussion to the Page 5 line 92 of the main text: “The MS, ¹H NMR, ¹³C NMR and UV-visible absorption and emission spectra of [Ru(bpy)₃]²⁺ were characterized in the supporting information (Supplementary Fig. 1).”

Supplementary Figure 1. Characterization of [Ru(bpy)₃]²⁺. (a) ¹H NMR spectra of [Ru(bpy)₃]²⁺. (b) ¹³C NMR spectra of [Ru(bpy)₃]²⁺. (c) UV-visible absorption (black) and emission (red) spectra of [Ru(bpy)₃]²⁺. (d) The MS spectra of [Ru(bpy)₃]²⁺.

2). Please investigate and discuss the reaction mechanism of US-activated $[Ru(bpy)_3]^{2+}$ for 1O₂ generation.

Reply to the reviewer and revisions made: We thank the reviewer for this advice. During the revision, we have updated the following discussion on Page 4 line 79: “The energy interval between LUMO and HOMO of $[Ru(bpy)_3]^{2+}$ is only 0.1239,³⁷ suggesting $[Ru(bpy)_3]^{2+}$ is easy to be excited into the highly oxidative excited-state species. Sharipov group have studied that the energy of part of sonoluminescence (300-452 nm) of water is high enough to excite $[Ru(bpy)_3]^{2+}$. Moreover, radical products of sonolysis of water can also excite $[Ru(bpy)_3]^{2+}$. The sonochemiluminescence spectra of $[Ru(bpy)_3]^{2+}$ was recorded by US irradiation of argon saturated aqueous solutions of $[Ru(bpy)_3]^{2+}$.³⁸⁻⁴⁰ In this work, we found that $[Ru(bpy)_3]^{2+}$ could be excited under US irradiation. In the presence of oxygen, US excited $[Ru(bpy)_3]^{2+}$ could transfer energy to oxygen to product ¹O₂.”

The result showed in the following is from the Ref[38-40]

ref[38]: Sonochemiluminescence in an aqueous solution of $Ru(bpy)_3Cl_2$

Formation of radical products

CL reactions of radicals and ions in solution

Radiative deactivation of the CL emitter

Scheme 1. Possible reactions leading to sonochemiluminescence of $[Ru(bpy)_3]^{2+}$.

37 Zheng, K. C.; Wang, J. P.; Peng, W. L.; Liu, X. W.; Yun, F. C., Theoretical Studies on the Electronic and Related Properties of $[Ru(L)_3]^{2+}$ (L=bpy, bpm, bpz) with DFT Method. *Mol. Struct.* **2002**, 582, 1-9.

38 Sharipov, G. L.; Abdrakhmanov, A. M.; Gareev, B. M.; Yakshembetova, L. R., Sonochemiluminescence in an Aqueous Solution of $Ru(bpy)_3Cl_2$. *Ultrason. Sonochem.* **2018**, 42, 526–531.

39 Sharipov, G. L.; Abdrakhmanov, A. M.; Yakshembetova, L. R., Mechanism of Multibubble Sonochemiluminescence of $Ru(bpy)_3^{2+}$ in Neutral Aqueous Solutions. *Ultrason. Sonochem.* **2019**,

51, 395–398.

40 Gareev, B. M.; Yakshembetova, L. R.; Abdrakhmanov, A. M.; Sharipov, G. L., Mechanism of the $\text{Ru}(\text{bpy})_3^{2+}$ Single-bubble Sonochemiluminescence in Neutral and Alkaline Aqueous Solutions. *J. Lumin.* **2019**, 208, 99-103.

3). *There are many mistakes in the manuscript. Line 110-111, ESR spectra of DMPO-OH is not Fig. 1b, and this data is missed by the authors in the manuscript. Line 129-130, the NADH depletion curves is not Fig. 2b. Line 178-179, the sentence 'To investigate the sonotherapy.....with SOSG and $[\text{Ru}(\text{bpy})_3]^{2+}$ ' is repeating. The authors should carefully check the manuscript and correct these mistakes.*

Reply to the reviewer and revisions made: We thank the reviewer for pointing these mistakes out. They have been corrected during the revision in text. ESR spectra of DMPO-OH is in Fig. 1a. The NADH depletion curves is in Fig. 2a. The sentence 'To investigate the sonotherapy.....with SOSG and $[\text{Ru}(\text{bpy})_3]^{2+}$ ' is deleted. We also carefully check the whole manuscript and correct mistakes.

4). *Please investigate and quantify the in vitro and in vivo NADH depletion by US-activated $[\text{Ru}(\text{bpy})_3]^{2+}$*

Reply to the reviewer: We have investigated and quantified the in vitro NADH depletion by US-activated $[\text{Ru}(\text{bpy})_3]^{2+}$ in Figure S12. "On the other hand, we further investigated NADH depletion in 4T1 cells (Supplementary Fig. 12). Under US irradiation, the intracellular NADH concentration reduced after incubation with $[\text{Ru}(\text{bpy})_3]^{2+}$, while only US irradiation or only $[\text{Ru}(\text{bpy})_3]^{2+}$ incubation, the NADH levels were unaffected." There is no NADH detection kit developed in vivo, thus we cannot detect the NADH concentration in vivo.

5). *In Figure 4b, there is lack of a control group that SOSG + $[\text{Ru}(\text{bpy})_3]^{2+}$ without US irradiation. Please add.*

Reply to the reviewer: We thank the reviewer for this suggestion. In Figure 4b, 0 min group is the control. The legend '0 min' was changed into 'without US'. And we adjust the fluorescent pseudo color.

6). *How does the author conduct the US treatment in the 'in vivo sonotherapy' section? What's the distance of the US probe with the tumor during the US treatment? The authors claim that the therapeutic effect can reach deep tissues over 10 cm under US irradiation. Thus, please add a deep tumor model to confirm the efficient SDT performance of $[\text{Ru}(\text{bpy})_3]^{2+}$.*

Reply to the reviewer and revisions made: We thank the reviewer for this advice. To confirm the efficient SDT in the deep-tumor tissues, we added a group of in vivo therapeutic experiment. In order

to enlarge the spread distance of US, the 4T1 tumor was transplanted on the right side of mice, and the US probe was on the left side of mice during treatment. The US wave penetrated from left to right. We added this data as Supplementary Figure 14. And we updated the following discussion on Page 10 line 210: “To confirm the efficient SDT in the deep-tumor tissues, the 4T1 tumor was transplanted on the right side of mice, and the US probe was on the left side of mice during treatment. The US wave penetrated from left to right. The result was shown in **Supplementary Fig. 14**, confirming that the efficient SDT of $[\text{Ru}(\text{bpy})_3]^{2+}$ can reach deep-tumor tissues.”

Supplementary Figure 14. *In vivo* therapeutic experiment in deep tumor model. (a) Schematic of the *in vivo* sonotherapy procedure in deep tumor model. The 4T1 tumor was transplanted on the right side of mice, and the US probe was on the left side of mice during treatment. The US waves penetrated from left side of mice to right side during SDT. (b) Photos of tumors were collected from mice at 14 day after various treatments. (c) Tumor growth curves of mice after various treatments. (d) Average tumor weights of mice at 14 day post various treatments.

7). *As for deep tumor model, systemic injection is more appropriate than local injection. Thus, can the authors achieve the SDT post i.v. injection of $[\text{Ru}(\text{bpy})_3]^{2+}$ with proper modification?*

Reply to the reviewer: We thank the reviewer for this suggestion. In our experiment, $[\text{Ru}(\text{bpy})_3]^{2+}$ could not accumulated in tumor tissue post i.v. injection. The bio-distribution of $[\text{Ru}(\text{bpy})_3]^{2+}$ was measured by collection main organs of mice 4 h after i.v. injection using ICP. High levels of Ru were detected in the kidney samples, indicating the elimination of $[\text{Ru}(\text{bpy})_3]^{2+}$ via the renal filtration pathway.

Our subsequent work will focus on SDT using $[\text{Ru}(\text{bpy})_3]^{2+}$ loaded on liposome or protein. We try to deliver $[\text{Ru}(\text{bpy})_3]^{2+}$ to tumor tissue by nano-carrier. And we updated the following discussion on conclusion part. “It is a pity that the Ru drug can only use intratumoral injection at the moment. In the future, our work will focus on delivering $[\text{Ru}(\text{bpy})_3]^{2+}$ to tumor tissue after systemic injection instead of local injection using $[\text{Ru}(\text{bpy})_3]^{2+}$ loaded on liposome or protein to improve tumor-targeting *in vivo*.”

8). How about the biosafety and the acute toxicity of $[\text{Ru}(\text{bpy})_3]^{2+}$? Some *in vivo* toxicity data should be provided at least to support the feasibility of this new material for biomedical applications.

Reply to the reviewer and revisions made: $[\text{Ru}(\text{bpy})_3]^{2+}$ is a common metal complex for PDT.[1-5] The $[\text{Ru}(\text{bpy})_3]^{2+}$ is biosafety and have been studied in many papers[1-5]. We added data about tissue sections of major organs collected from mice i.v. injected with five times of the therapeutic dose of $[\text{Ru}(\text{bpy})_3]^{2+}$ in Supplementary Figure 15. The results showed that no obvious tissue damage was found from these slices. We further studied that the LD_{50} of $[\text{Ru}(\text{bpy})_3]^{2+}$ was 3.89 mg kg^{-1} in the acute toxicity experiment. These results mean that $[\text{Ru}(\text{bpy})_3]^{2+}$ is safe *in vivo*.

Supplementary Figure 15. The biosafety and the acute toxicity of $[\text{Ru}(\text{bpy})_3]^{2+}$. Healthy mice were i.v. injected with 2.5 mg kg^{-1} $[\text{Ru}(\text{bpy})_3]^{2+}$. Mice were sacrificed at 1 or 7 day post injection to collect their main organ for H&E stained slices. No obvious tissue damage can be found from these

slices. In acute toxicity experiment of $[\text{Ru}(\text{bpy})_3]^{2+}$, each i.v. injection dose was investigated in six mice. The LD_{50} of $[\text{Ru}(\text{bpy})_3]^{2+}$ was 3.89 mg kg^{-1} .

Ref

- [1] Taurine-modified Ru (II)-complex targets cancerous brain cells for photodynamic therapy. *Chemical Communications* 53.44 (2017): 6033-6036.
- [2] Ruthenium (II) polypyridyl complexes as mitochondria-targeted two-photon photodynamic anticancer agents. *Biomaterials* 56 (2015): 140-153.
- [3] Nanoscale metal-organic layers for deeply penetrating X-ray-induced photodynamic therapy. *Angewandte Chemie* 129.40 (2017): 12270-12274.
- [4] Ruthenium (II) complex incorporated UiO-67 metal-organic framework nanoparticles for enhanced two-photon fluorescence imaging and photodynamic cancer therapy. *ACS Appl. Mater. Interfaces* 9 (2017) 5699-5708.
- [5] Visible light photocatalysis as a greener approach to photochemical synthesis. *Nat. Chem.* 2 (2010) 527.

Reply to Reviewer 2:

The authors present an interesting paper about the sonosensitizing properties of Ruthenium(II) providing the results of its activation under US exposure on in vitro and in vivo models. The manuscript could be improved by considering the following revisions.

Reply to the reviewer: We thank the reviewer for the enthusiastic comments. Our responses to these issues are attached as follows.

1). The authors performed the in vitro sonodynamic treatment after a cell incubation of 4 h with Ruthenium(II) but a Ruthenium(II) cell uptake study was not provided. How did the authors select the Ruthenium(II) time of cell incubation for the sonodynamic experiments?

Reply to the reviewer: We thank the reviewer for this comment. A $[\text{Ru}(\text{bpy})_3]^{2+}$ cell uptake study was added as **Supplementary Fig. 11**. We chose 4 h incubation time of $[\text{Ru}(\text{bpy})_3]^{2+}$ because it was well uptake by cells after 4 h.

Supplementary Figure 11. The cell uptake of $[\text{Ru}(\text{bpy})_3]^{2+}$ by 4T1 cells for different incubation time.

2). The authors performed the in vivo sonodynamic treatment by an intratumoral administration of

Ruthenium(II). Do the authors know the Ruthenium(II) biodistribution after e.v. or i.p. administration? Are there data about the systemic toxicity of Ruthenium(II)?

Reply to the reviewer and revisions made: We thank the reviewer for this suggestion. We found that $[\text{Ru}(\text{bpy})_3]^{2+}$ could not accumulated in tumor tissue post i.v. injection. The bio-distribution of $[\text{Ru}(\text{bpy})_3]^{2+}$ was measured by collection main organs of mice 4 h after i.v. injection using ICP. High levels of Ru were detected in the kidney samples, indicating the elimination of $[\text{Ru}(\text{bpy})_3]^{2+}$ via the renal filtration pathway. Our subsequent work will focus on SDT using $[\text{Ru}(\text{bpy})_3]^{2+}$ loaded on liposome or protein. We try to deliver $[\text{Ru}(\text{bpy})_3]^{2+}$ to tumor tissue by nano-carrier. And we updated the following discussion on conclusion part. “It is a pity that the Ru drug can only use intratumoral injection at the moment. In the future, our work will focus on delivering $[\text{Ru}(\text{bpy})_3]^{2+}$ to tumor tissue after systemic injection instead of local injection using $[\text{Ru}(\text{bpy})_3]^{2+}$ loaded on liposome or protein to improve tumor-targeting *in vivo*.”

We added data about tissue sections of major organs collected from mice i.v. injected with five times of the therapeutic dose of $[\text{Ru}(\text{bpy})_3]^{2+}$ in Supplementary Figure 15. The results showed that no obvious tissue damage was found from these slices. We further studied that the LD_{50} of $[\text{Ru}(\text{bpy})_3]^{2+}$ was 3.89 mg kg^{-1} in the acute toxicity experiment. These results mean that $[\text{Ru}(\text{bpy})_3]^{2+}$ is safe *in vivo*.

Supplementary Figure 15. The biosafety and the acute toxicity of $[\text{Ru}(\text{bpy})_3]^{2+}$. Healthy mice were i.v. injected with $2.5 \text{ mg kg}^{-1} [\text{Ru}(\text{bpy})_3]^{2+}$. Mice were sacrificed at 1 or 7 day post injection to collect their main organ for H&E stained slices. No obvious tissue damage can be found from these slices. In acute toxicity experiment of $[\text{Ru}(\text{bpy})_3]^{2+}$, each i.v. injection dose was investigated in six mice. The LD_{50} of $[\text{Ru}(\text{bpy})_3]^{2+}$ was 3.89 mg kg^{-1} .

3). The authors used high US frequency (3 MHz) and low US intensity (0.3 W cm^{-2}) for the insonation compared to the commonly used US parameters for in vivo sonodynamic treatment. Why did the authors select these US parameters?

Reply to the reviewer: We thank the reviewer for this comment. Our US instrument only supports stationary US frequency 3MHz. The US power of 0.3 W cm^{-2} was selected due to the temperature increase of the solution under higher US power ($>0.3 \text{ W cm}^{-2}$). For example, US irradiation with the power of 0.4 W cm^{-2} showed obvious heating effect on aqueous solution, and the temperature was high enough to kill 4T1 tumor cells directly (Supplementary Fig. 10a-c).

Supplementary Figure 10. The temperature change during US irradiation. (a) The temperature curve of $[\text{Ru}(\text{bpy})_3]^{2+}$ solution during 0.3 W cm^{-2} US irradiation. (b) The temperature curve of $[\text{Ru}(\text{bpy})_3]^{2+}$ solution during 0.4 W cm^{-2} US irradiation. (c) The cell viabilities of 4T1 cells after incubation with different US powers for 20 min. (d) The temperature curve of tumor area during

SDT in (e). (e) Near infrared thermal imaging of anesthetized mice during US irradiation (0.3 W cm⁻², 3 MHz, 20 min).

4). Which is the mechanism underpinning the Ruthenium(II) activation? Do the authors suppose the occurrence of acoustic cavitation or sonoluminescence? According to literature, the US frequency and intensity used by the authors are not so efficient in the generation of acoustic cavitation and sonoluminescence. The authors have to explain their hypothesis about the Ruthenium(II) activation by US in the discussion section, explaining the reason why they selected the US parameters used.

Reply to the reviewer and revisions made: We thank the reviewer for this advice. According to literature, the mechanism underpinning the [Ru(bpy)₃]³⁺ activation was both acoustic cavitation and sonoluminescence. During the revision, we have updated the following discussion on Page 4 line 79: “In this work, we found that [Ru(bpy)₃]²⁺ could be excited under US irradiation. The energy interval between LUMO and HOMO of [Ru(bpy)₃]²⁺ is only 0.1239,³⁷ suggesting [Ru(bpy)₃]²⁺ is easy to be excited into the highly oxidative excited-state species. Sharipov group have studied that the energy of part of sonoluminescence (300-452 nm) of water is high enough to excite [Ru(bpy)₃]²⁺. Moreover, radical products of sonolysis of water can also excite [Ru(bpy)₃]²⁺. The sonochemiluminescence spectra of [Ru(bpy)₃]²⁺ was recorded by US irradiation of argon saturated aqueous solutions of [Ru(bpy)₃]²⁺.³⁸⁻⁴⁰ In the presence of oxygen, US excited [Ru(bpy)₃]²⁺ could transfer energy to oxygen to product ¹O₂.”

[38] Sonochemiluminescence in an aqueous solution of Ru(bpy)₃Cl₂

Formation of radical products

CL reactions of radicals and ions in solution

Radiative deactivation of the CL emitter

Scheme 1. Possible reactions leading to sonochemiluminescence of [Ru(bpy)₃]²⁺.

[38] Sonochemiluminescence in an aqueous solution of Ru(bpy)₃Cl₂

Our US instrument only supports stationary US frequency 3MHz. The US power of 0.3 W cm⁻² was selected due to the temperature increase of the solution under higher US power (>0.3 W cm⁻²). For example, US irradiation with the power of 0.4 W cm⁻² showed obvious heating effect on aqueous

solution, and the temperature was high enough to kill 4T1 tumor cells directly (Supplementary Fig. 10a-c).

5). About the Ruthenium(II) generation of 1O_2 and $\bullet OH$ under US exposure, it would be very interesting to know what happens during US exposure alone (the ESR spectra of US alone are not shown in Figure 1 or in supplementary materials).

Reply to the reviewer: We thank the reviewer for this advice. The ESR spectra of water solvents alone by US irradiation were studied as Supplementary Fig. 2. We did not find obvious 1O_2 and $\bullet OH$ generation in water solution under US irradiation. Thus, $[Ru(bpy)_3]^{2+}$ excited by radical products of sonolysis of water was not the dominant mechanism of $[Ru(bpy)_3]^{2+}$ excited by US.

Supplementary Figure 2. The ESR spectra of US irradiation alone without $[Ru(bpy)_3]^{2+}$ in water. The TEMP and DMPO were used as 1O_2 and $\bullet OH$ trapping agents, respectively.

6). What about the medium temperature during the ESR experiments that are performed by a long time of US exposure?

Reply to the reviewer and revisions made: We thank the reviewer for this question. During the ESR experiments, the medium temperature rise slightly with 2 h of $0.3W\text{ cm}^{-2}$ US irradiation. This data was added into Supplementary Figure 10a.

Supplementary Figure 10. (a) The temperature curve of $[\text{Ru}(\text{bpy})_3]^{2+}$ solution during 2 h 0.3 W cm^{-2} US irradiation.

7). In Figure 3a and b the statistical significance has to be shown.

Reply to the reviewer and revisions made: We thank the reviewer for this advice. The statistical significance has to be added in Figure 3a and b.

8). There are some oversights throughout the manuscript.

Reply to the reviewer: We thank the reviewer for this advice. We would carefully check the manuscript and correct mistakes.

Reply to Reviewer 3:

In this article, the author pioneered the use of $[Ru(bpy)_3]^{2+}$ as a kind of sonosensitizer and tried to demonstrate that $[Ru(bpy)_3]^{2+}$ -mediated sonodynamic therapy effectively killed the breast cancer cells in vitro and in vivo. The mechanism of cell death is mainly due to the generation of singlet oxygen promoted by sonodynamic therapy. The part of physics is well organized, but the evidence in biological section is obviously insufficient. This research just demonstrated the cell-killing effect of a therapy and its ability to produce singlet oxygen. Otherwise, the following questions puzzled me a lot. I think this study needs to be carefully planned again to promote the innovation.

Reply to the reviewer: We thank the reviewer for the comments. Our responses to these issues are attached as follows. During the revision, we will try our best to add evidence in biological section per your following questions.

1). According to the instructions of SOSG (Invitrogen, S36002), SOSG is cell-impermeant thus it's not proper to be used to test the singlet oxygen in cells or tissues. If your SOSG is available, please provide the brand and catalog number of the drug.

Reply to the reviewer: We thank the reviewer for this comment. Our SOSG is obtained from Invitrogen, its catalog number is S36002. But we think that SOSG can spread into 4T1 cells, some references also showed that SOSG could be uptake by living cancer cells (refs[1-4] in the following)^[1-4]. To explore this problem, we designed the following experiment. 4T1 cells were incubated with Ce6 (a high singlet oxygen yield photosensitizer) for 4 h and then removed Ce6. After that 4T1 cells were incubated with SOSG for 0.5 or 4 h. SOSG was removed before 660 nm laser irradiation. The results showed that endocytosed SOSG can react with singlet oxygen and observe visible green fluorescence in living cells. The fluorescence intensity did not change significantly under longer incubation time of SOSG. This experiment suggested that SOSG could spread into living 4T1 cells even without assisted by singlet oxygen.

The same results could be found in $[Ru(bpy)_3]^{2+}$ based SDT. We double checked our data about SOSG, and we confirm SOSG could work as well as DCFH-DA (Figure S12). If the reviewer think this data is controversial, we can replace Figure 3d with Supplementary Figure 12.

According to literature, SOSG indeed entered living 4T1 tumor cells, Hela cells, and HepG2 tumor cells.

Figure S16. CLSM analysis of intracellular singlet oxygen in 4T1 cells treated by three formulations (Ce6, Ce6@CMOF and Ce6@RMOF) with and without light treatment (660 nm, 20 mW cm⁻², 10 min). The specific fluorescence probe (SOSG) was used for the imaging. Scale bar: 20 μm.

[2] Triggered all-active metal organic framework: Ferroptosis machinery contributes to the apoptotic photodynamic antitumor therapy. *Nano Lett.* 10.1021/acs.nanolett.9b02904.

Figure 6. (a) CLSM images of DANO-treated HeLa cells after LED light irradiation (λ 405 nm, 20 mW cm⁻², t_{irr} = 0, 1 and min). The cells were stained with Mito Red Tracker. Scale bars, 10 μm. (b) The median fluorescence intensities derived from images in a. Statistical significance was assessed using a two-way analysis of variance test (** $P \leq 0.05$, *** $P \leq 0.001$). (c) Fluorescence images of intracellular RONS. Scale bars, 50 μm.

[3] Cascade Reactions by Nitric Oxide and Hydrogen Radical for Anti-Hypoxia Photodynamic Therapy Using an Activatable Photosensitizer. *J. Am. Chem. Soc.* Doi: 10.1021/jacs.0c10517.

Figure 3. (a) Schematic illustration of DHE for $O_2^{\cdot-}$ detection. (b) ROS generation of ENBS-B in HepG2 cells upon 660 nm light irradiation and CLSM images of cellular $O_2^{\cdot-}$ after exposure to 14.4 J/cm² light dose in the absence or presence of Vc. (c) Intracellular hypoxia imaging using ROS-ID as an anaerobic indicator. (d) ROS detection in HepG2 cells under normoxia (21% O_2) and hypoxia (2% O_2) conditions using DHE, SOSG, and HPF as the $O_2^{\cdot-}$, 1O_2 , and OH^{\cdot} fluorescence indicator, respectively. SOD inhibitor-mediated cellular (e) $O_2^{\cdot-}$ and (f) OH^{\cdot} generation. * $P < 0.05$, *** $P < 0.001$, and **** $P < 0.0001$ determined by Student's t test.

[4] Near-infrared light-initiated molecular superoxide radical generator: rejuvenating photodynamic therapy against hypoxic tumors. *J. Am. Chem. Soc.* 2018, 140, 14851–14859

Refs in this part:

[1] Singlet Oxygen Sensor Green®: Photochemical Behavior in Solution and in a Mammalian Cell. *Photochem. Photobiol.* 2011 May-Jun; 87 (3):671-679.

[2] Triggered all-active metal organic framework: Ferroptosis machinery contributes to the apoptotic photodynamic antitumor therapy. *Nano Lett.* 10.1021/acs.nanolett.9b02904

[3] Cascade Reactions by Nitric Oxide and Hydrogen Radical for Anti-Hypoxia Photodynamic Therapy Using an Activatable Photosensitizer. *J. Am. Chem. Soc.* Doi: 10.1021/jacs.0c10517.

[4] Near-infrared light-initiated molecular superoxide radical generator: rejuvenating photodynamic therapy against hypoxic tumors. *J. Am. Chem. Soc.* 2018, 140, 14851–14859

2). I wonder the basis for determining the duration and manner of $[Ru(bpy)_3]^{2+}$ injection. Why you choose to expose tumors to US just at 10 min post injection of $[Ru(bpy)_3]^{2+}$? No direct proof of $[Ru(bpy)_3]^{2+}$ distribution within the tumor can be found in this passage. And why you choose the manner of intrathecal injection but not intravenous injection or intraperitoneal injection? If you have strong evidence, please mark the references.

Reply to the reviewer: We thank the reviewer for this suggestion. We have tracked $[Ru(bpy)_3]^{2+}$ by in vivo fluorescence imaging after i.v. administration of $[Ru(bpy)_3]^{2+}$. It was found that $[Ru(bpy)_3]^{2+}$ has no tumor targeting ability. The bio-distribution of $[Ru(bpy)_3]^{2+}$ was measured by collection of main organs of mice 4 h after i.v. injection using ICP. High levels of Ru were detected in the kidney samples, indicating the elimination of $[Ru(bpy)_3]^{2+}$ via the renal filtration pathway. Thus, we can only inject $[Ru(bpy)_3]^{2+}$ directly into the tumor tissue, so that tumor tissue could obtain sufficient concentration $[Ru(bpy)_3]^{2+}$ during SDT. We carried out SDT at 10 min post injection of $[Ru(bpy)_3]^{2+}$, only because we need 10 min to stanch bleeding, fix mice and apply ultrasonic glue before SDT. The

data of $[\text{Ru}(\text{bpy})_3]^{2+}$ distribution within the tumor is inaccurate by in vivo fluorescence imaging, because the emitted light of $[\text{Ru}(\text{bpy})_3]^{2+}$ away from near infrared imaging window. All manners of $[\text{Ru}(\text{bpy})_3]^{2+}$ injection are within the safe dose.

$[\text{Ru}(\text{bpy})_3]^{2+}$ is a common metal complex for PDT.[1-5] The $[\text{Ru}(\text{bpy})_3]^{2+}$ is biosafety and have been studied in many papers[1-5]. We added data about tissue sections of major organs collected from mice i.v. injected with five times of the therapeutic dose of $[\text{Ru}(\text{bpy})_3]^{2+}$ in Supplementary Figure 15. The results showed that no obvious tissue damage was found from these slices. We further studied that the LD_{50} of $[\text{Ru}(\text{bpy})_3]^{2+}$ was 3.89 mg kg^{-1} in the acute toxicity experiment. These results mean that $[\text{Ru}(\text{bpy})_3]^{2+}$ is safe in vivo.

Supplementary Figure 15. The biosafety and the acute toxicity of $[\text{Ru}(\text{bpy})_3]^{2+}$. Healthy mice were i.v. injected with 2.5 mg kg^{-1} $[\text{Ru}(\text{bpy})_3]^{2+}$. Mice were sacrificed at 1 or 7 day post injection to collect their main organ for H&E stained slices. No obvious tissue damage can be found from these slices. In acute toxicity experiment of $[\text{Ru}(\text{bpy})_3]^{2+}$, each i.v. injection dose was investigated in six mice. The LD_{50} of $[\text{Ru}(\text{bpy})_3]^{2+}$ was 3.89 mg kg^{-1} .

Ref

[1] Taurine-modified Ru (II)-complex targets cancerous brain cells for photodynamic therapy.

Chemical Communications 53.44 (2017): 6033-6036.

[2] Ruthenium (II) polypyridyl complexes as mitochondria-targeted two-photon photodynamic anticancer agents. *Biomaterials* 56 (2015): 140-153.

[3] Nanoscale metal-organic layers for deeply penetrating X-ray-induced photodynamic therapy. *Angewandte Chemie* 129.40 (2017): 12270-12274.

[4] Ruthenium (II) complex incorporated UiO-67 metal-organic framework nanoparticles for enhanced two-photon fluorescence imaging and photodynamic cancer therapy. *ACS Appl. Mater. Interfaces* 9 (2017) 5699-5708.

[5] Visible light photocatalysis as a greener approach to photochemical synthesis. *Nat. Chem.* 2 (2010) 527.

3). According to Figure 3b, why is the duration of treatment determined at 20 min instead of 15 min? It's obvious that US for 20 min can cause a certain percentage of cell death.

Reply to the reviewer: We thank the reviewer for this suggestion. From Figure 3b, US irradiation for 20 min just cause a little cell death, and US irradiation for 25 min can cause a certain percentage of cell death. Thus we think 20 min is a better choose.

4). The reactive oxygen species other than singlet oxygen and hydroxyl radicals were not excluded in this study.

Reply to the reviewer: We thank the reviewer for the comments. Generally speaking, SDT related studies only consider the production of singlet oxygen and hydroxyl radicals. According to studies of $[\text{Ru}(\text{bpy})_3]^{2+}$ based PDT, excited $[\text{Ru}(\text{bpy})_3]^{2+}$ tend to produce singlet oxygen. Singlet oxygen and hydroxyl radicals are low energy, and are easy to produce, compared to other reactive oxygen species.

5). There is no sufficient evidence to prove the connection between cell death and singlet oxygen generation except for one experiment on 4T1 cells using NaN_3 .

Reply to the reviewer: Generally speaking, intracellular singlet oxygen show a strong killing effect of tumor cells. We have detected the singlet oxygen in 4T1 tumor cells in the present of $[\text{Ru}(\text{bpy})_3]^{2+}$ during US irradiation. Thus, we try to add NaN_3 during *in vitro* sonotherapy experiments. Compared with figure 3a, adding NaN_3 , sono-cytotoxicity of $[\text{Ru}(\text{bpy})_3]^{2+}$ reduce and IC_{50} of $[\text{Ru}(\text{bpy})_3]^{2+}$ increase from 2.91 μM to 5.85 μM . Since singlet oxygen was eliminated by NaN_3 , US excited $[\text{Ru}(\text{bpy})_3]^{2+}$ still oxidized intracellular NADH, leading to the death of 4T1 tumor cells.

Moreover, we tried to detect superoxide radical by using DHE probe and hydroxyl radical by using HPF probe. From the results, hydroxyl radical and superoxide radical could not be detected during US irradiation.

6). There are some mistakes in grammar and diction in the article as follow:

- 1) In Page3 Line55 “intensit” should be changed to “intensity”;
- 2) In Page3 Line57-58 “Sonodynamic therapy (SDT) is a novel non-invasive strategy, which triggered by the high tissue penetration ultrasound (up to 10 cm)” should be changed to “Sonodynamic therapy (SDT) is a novel non-invasive strategy, which is triggered by the high tissue penetration ultrasound (up to 10 cm)”;
- 3) In Page3 Line64-66 “However, organic sonosensitizers often exhibit limited stability under US, and some of complicated nanoparticles show low ROS quantum yield decreasing SDT efficiency” should be changed to “However, organic sonosensitizers often exhibit limited stability under US, and some of complicated nanoparticles show low ROS quantum yield thus decreasing SDT efficiency”;
- 4) In Page4 Line70-72 “NADH is produced by glycolysis process and citric acid cycle in cellular respiration, and considers as the carrier of biological hydrogen and electron donor in living cells” should be changed to “NADH is produced by glycolysis process and citric acid cycle in cellular respiration, and is considered as the carrier of biological hydrogen and electron donor in living

cells”;

5) In Page4 Line74, it is not proper to describe PDT as a kind of cell damage, PDT is a kind of therapy.

6) In Page6 Line117-119 “The results showed that $[Ru(bpy)_3]^{2+}$ exhibited a strong ability to produce $1 O_2$ under US and high sono-stability, suggesting that $[Ru(bpy)_3]^{2+}$ has potential to be an excellent sono-sensitizer, and its probable mechanism was shown in Fig. 1d” should be changed to “The results showed that $[Ru(bpy)_3]^{2+}$ exhibited strong ability to produce $1 O_2$ and high sono-stability under US, suggesting that $[Ru(bpy)_3]^{2+}$ has the potential to be an excellent sono-sensitizer, and the probable mechanism was shown in Fig. 1d”;

7) In Page7 Line137-139 “The results suggested that NADH transforming into $NADH^{\bullet}$ via single electron transfer mechanism by transferring an electron to US induced excited-state $[Ru(bpy)_3]^{2+}$ ” should be changed to “The results suggested that NADH was transformed into $NADH^{\bullet}$ via single electron transfer mechanism by transferring an electron to $[Ru(bpy)_3]^{2+}$ which is excited by US”;

8) In Page7 Line143-146 “Then, irradiated with US ($0.3 W cm^{-2}$, 3.0 MHz) for 20 min, NADH was converted into its oxidized form NAD^+ (with new peaks at 8.31, 8.55, 8.99, 9.36 and 9.58 ppm assignable to the hydrogen atoms of NAD^+) was observed in the $^1 H$ NMR spectrum” should be changed to “Then, after being irradiated with US ($0.3 W cm^{-2}$, 3.0 MHz) for 20 min, NADH was converted into its oxidized form NAD^+ (with new peaks at 8.31, 8.55, 8.99, 9.36 and 9.58 ppm assignable to the hydrogen atoms of NAD^+) which can be observed in the $^1 H$ NMR spectrum”;

9) In Page10 Line209-210, “Average tumor weight in $[Ru(bpy)_3]^{2+} + US$ group was the least in all four groups and even a mouse was cure” should be changed to “Among the four groups, the average tumor weight in $[Ru(bpy)_3]^{2+} + US$ group was the least and one of the five mice was completely cured”.

Reply to the reviewer: We thank the reviewer very much for pointing these mistakes out. They have been corrected during the revision and we have carefully checked the manuscript again.

Reviewers' Comments:

Reviewer #1:

Remarks to the Author:

This work after revisions is now acceptable for publication.

Reviewer #2:

Remarks to the Author:

The authors have addressed my concerns and in my opinion the manuscript is now worth to be published in Nature Communications.

Reviewer #3:

Remarks to the Author:

In this article, the author pioneered the use of $[\text{Ru}(\text{bpy})_3]^{2+}$ as a kind of sonosensitizer and tried to demonstrate that $[\text{Ru}(\text{bpy})_3]^{2+}$ -mediated sonodynamic therapy effectively killed the breast cancer cells in vitro and in vivo. However, the experimental design was not rigorous enough, we can not find direct linkage between singlet oxygen generation, NADH oxidation and cell death. And the first use of $[\text{Ru}(\text{bpy})_3]^{2+}$ as sonosensitizer requires full optimization of ultrasonic parameters and medication methods. The US frequency and intensity used in both cell and animal experiments need to be adequately demonstrated.

From the innovative point of view, both singlet oxygen generation and strong penetration of SDT are well known. The biological effects of NADH oxidation are interesting but have not been discussed in depth, at least not directly related to cytotoxicity of SDT. The study was not as innovative as the journal's usual level. The manuscript still need to be improved by considering the following revisions.

1. According to the reply, the basis for determining the duration between $[\text{Ru}(\text{bpy})_3]^{2+}$ injection and US irradiation is not scientific enough. From the perspective of SDT, you at least need to ensure the distribution of $[\text{Ru}(\text{bpy})_3]^{2+}$ in cells of the tumor, otherwise it wouldn't be called SDT. The results of singlet oxygen generation in tissues can not prove whether the drug is inside the cells or the extracellular matrix. According to Supplementary Figure 11, it takes 4 h for 4T1 cells to fully absorb $[\text{Ru}(\text{bpy})_3]^{2+}$ in vitro, this further suggests that 10 min is far from enough.
2. Are you sure the abscissa of Figure 3b is "Concentration" or "Time"? Please check it carefully.
3. The generation of singlet oxygen is not equal to cell death. To prove that cell death is directly related to singlet oxygen production, you at least need to add a group of "Ru+ NaN_3 +US" in Figure 3c and Figure 5f.
4. The time points corresponding to the different color curves are not shown in Supplementary Figure 4.
5. In this passage, the method of determining the ultrasonic parameters is not scientific enough, the parameters need to be further optimized.

Dear Editor,

We would like to thank all the reviewers for their insightful comments about our work again. The following are our point-to-point responses to the concerns raised and revisions made accordingly.

Reply to Reviewer 1:

This work after revisions is now acceptable for publication.

Reply to the reviewer: We thank the reviewer for the recognition of this work.

Reply to Reviewer 2:

The authors have addressed my concerns and in my opinion the manuscript is now worth to be published in Nature Communications.

Reply to the reviewer: We thank the reviewer for the recognition of this work.

Reply to Reviewer 3:

In this article, the author pioneered the use of $[Ru(bpy)_3]^{2+}$ as a kind of sonosensitizer and tried to demonstrate that $[Ru(bpy)_3]^{2+}$ -mediated sonodynamic therapy effectively killed the breast cancer cells in vitro and in vivo. However, the experimental design was not rigorous enough, we can not find direct linkage between singlet oxygen generation, NADH oxidation and cell death. And the first use of $[Ru(bpy)_3]^{2+}$ as sonosensitizer requires full optimization of ultrasonic parameters and medication methods. The US frequency and intensity used in both cell and animal experiments need to be adequately demonstrated. From the innovative point of view, both singlet oxygen generation and strong penetration of SDT are well known. The biological effects of NADH oxidation are interesting but have not been discussed in depth, at least not directly related to cytotoxicity of SDT. The study was not as innovative as the journal's usual level. The manuscript still need to be improved by considering the following revisions.

Reply to the reviewer: We thank the reviewer for these meticulous comments. Our responses to these issues are attached as follows.

1). *According to the reply, the basis for determining the duration between $[Ru(bpy)_3]^{2+}$ injection and US irradiation is not scientific enough. From the perspective of SDT, you at least need to ensure the distribution of $[Ru(bpy)_3]^{2+}$ in cells of the tumor, otherwise it wouldn't be called SDT. The results of singlet oxygen generation in tissues can not prove whether the drug is inside the cells or*

the extracellular matrix. According to Supplementary Figure 11, it takes 4 h for 4T1 cells to fully absorb $[\text{Ru}(\text{bpy})_3]^{2+}$ in vitro, this further suggests that 10 min is far from enough.

Reply to the reviewer: We thank the reviewer for this suggestion.

Follow your suggestion, we tried to investigate the distribution of $[\text{Ru}(\text{bpy})_3]^{2+}$ after injection. We found that $[\text{Ru}(\text{bpy})_3]^{2+}$ was indeed in the extracellular matrix before 4 h injection and $[\text{Ru}(\text{bpy})_3]^{2+}$ spread inside the cells after 4 h injection.

As shown in the following images, the black cavity in the red circle was intracellular area. 10 min, 1 h, and 2 h post injection, the fluorescence signals of cell membrane and $[\text{Ru}(\text{bpy})_3]^{2+}$ were nearly overlap, which meant $[\text{Ru}(\text{bpy})_3]^{2+}$ was still in the extracellular matrix. However, after 4 h injection, the signal of $[\text{Ru}(\text{bpy})_3]^{2+}$ filled in the red circle, which meant $[\text{Ru}(\text{bpy})_3]^{2+}$ was spread inside the tumor cells.

The fluorescence images of DiD (red) co-stained tumor slices collected from mice after $[\text{Ru}(\text{bpy})_3]^{2+}$ injection. DiD and $[\text{Ru}(\text{bpy})_3]^{2+}$ were excited at 640 and 440 nm, respectively. Cell membrane dye: DiD (1,1-dioctadecyl-3,3,3,3-tetramethylindotricarbocyanine iodide).

Next, we improve the experimental scheme and carry out a new round of in vivo SDT experiments after 4 h injection. The different intensities of US irradiation used in both cells and animal experiments were studied as well. As shown in Fig. 5 and Supplementary Fig. 9b and Fig. 10, the US power of 0.3 W cm^{-2} was best selected due to the temperature increase of the solution under higher

US power ($>0.3 \text{ W cm}^{-2}$). For example, US irradiation with the power of 0.4 W cm^{-2} showed obvious heating effect on aqueous solution, and the temperature was high enough to kill 4T1 tumor cells directly (Supplementary Fig. 10). In addition, the tumor killing efficiency decrease under lower power (0.1 W cm^{-2} and 0.2 W cm^{-2}) of US irradiation (Supplementary Fig. 9b and Fig. 5).

Figure 5. *In vivo* sonotherapy. (a) Schematic of the *in vivo* sonotherapy procedure in 4T1 tumor bearing mice. Mice was irradiated by US ($0.1, 0.2, 0.3 \text{ W cm}^{-2}$, 3 MHz) for 20 min after 4 h i.t. injected with $[\text{Ru}(\text{bpy})_3]^{2+}$ solution. Tumor sizes were monitored every two days for 14 days in total. (b) Tumor growth curves of mice after various treatments. Error bars were standard errors based on five mice in each group ($***p < 0.001$, $**p < 0.01$, or $*p < 0.05$). (c) Representative images of mice at day 14 after various treatments. (d) Average tumor weights of mice at 14 day post various

treatments as shown in (e). (e) Photos of tumors were collected from mice at 14 day after various treatments.

2). Are you sure the abscissa of Figure 3b is “Concentration ” or “Time”? Please check it carefully

Reply to the reviewer: We thank the reviewer for pointing the mistake out. It has been corrected during the revision. The abscissa of Figure 3b is “Time”.

3). The generation of singlet oxygen is not equal to cell death. To prove that cell death is directly related to singlet oxygen production, you at least need to add a group of “Ru+ NaN₃+US” in Figure 3c and Figure 5f.

Reply to the reviewer: We thank the reviewer for this suggestion. We added a group of “Ru+ NaN₃+US” in Figure 3c and Figure 6a. And we updated the following discussion in the Results part. “The cytotoxicity of [Ru(bpy)₃]²⁺ for sonotherapy on 4T1 cells was inhibited by adding NaN₃ (a ¹O₂ scavenger) (Fig. 3c).” and “The tumor tissue in NaN₃ and [Ru(bpy)₃]²⁺ added with US irradiation group showed decrease tissue damage compared to [Ru(bpy)₃]²⁺ and US treated group, but obviously tissue damage compared to the other three groups. The results suggested that SDT damage to tumor tissue was related to singlet oxygen generation (Fig. 6a).”

Figure 3c

Figure 6a

4). The time points corresponding to the different color curves are not shown in Supplementary Figure 4.

Reply to the reviewer: We thank the reviewer for pointing these mistakes out. It has been corrected during the revision.

5). In this passage, the method of determining the ultrasonic parameters is not scientific enough, the parameters need to be further optimized.

Reply to the reviewer: Generally, the power of diagnostic ultrasound in clinic is 3~3.5 MHz. There are only two fixed frequencies of DJO-2776 sonicator (1 MHz and 3 MHz). These two fixed frequencies are two kinds of safe power widely recognized in clinic. 1 MHz and 3 MHz are used in most SDT-related researches.^[1] In our research, we found that there is no significant difference in efficiency of singlet oxygen generation and NADH oxidation. During the revision, we added ultrasonic power gradient experiments *in vitro* and *in vivo* (Fig. 5 and Supplementary Fig. 9b and Fig. 10). The results showed that the US power of 0.3 W cm⁻² was best selected due to the temperature increase of the solution under higher US power (>0.3 W cm⁻²). For example, US irradiation with the power of 0.4 W cm⁻² showed obvious heating effect on aqueous solution, and the temperature was high enough to kill 4T1 tumor cells directly (Supplementary Fig. 10). In addition, the tumor killing efficiency decrease under lower power (0.1 W cm⁻² and 0.2 W cm⁻²) of US irradiation (Supplementary Fig. 9b and Fig. 5).

Treatment	Materials	In vitro test	In vivo test	Power	Ref.	
SDT	TTP	MCF-7	MCF-7	1 MHz, 1 W cm ⁻²	6	
	HMME	MDA-MB-231/A-549	MDA-MB-231	1 MHz, 1.5 W cm ⁻²	7	
	HMME	HUVECs	MDA-MB-231	0.98 MHz, 1286 W cm ⁻²	8	
	HMME	4T1	4T1	3 MHz, 5 W cm ⁻²	9	
	Pplx	HeLa	HeLa	1 MHz, 1.5 W cm ⁻²	10	
	IR780	4T1	4T1	650 kHz, 2.4 W cm ⁻²	11	
	ICG	Hep1-6/LM3	HCC	1 MHz, 1.61 W cm ⁻²	12	
	APHB	HeLa	4T1	1 MHz, 0.6 W cm ⁻²	13	
	Berberine	HeLa	HeLa	1 MHz, 1 W cm ⁻²	14	
	TiO ₂	SCC7/NIH3T3	SCC7	1 MHz, 30 W	15	
	TiO ₂	HepG2	HepG2	0.8 MHz, 1.5 W cm ⁻²	16	
	MnWO _x	4T1	4T1	40 kHz, 3 W cm ⁻²	17	
	IR780	PANC-1	PANC-1	1 MHz, 1 W cm ⁻²	18	
	PpIX	U87	U87	1 MHz, 1.5 W cm ⁻²	19	
	PMCS	4T1	4T1	1 MHz, 1.5 W cm ⁻²	20	
	BP	4T1/L929/HeLa	4T1	1 MHz, 1 W cm ⁻²	21	
	SDT + PDT	HMME + Fe ₃ O ₄	T24/MRSA/ E. coli	ND	2 W cm ⁻²	23
		HP + ICG	RIF-1	RIF-1	1 MHz, 2.5 W cm ⁻²	24
		Rose Bengal	HepG2	ND	1 MHz, 2 W cm ⁻²	25
		Sonnelux	EAC	EAC	0.8 MHz, 0.5-3 W cm ⁻²	26
		Sonoflora 1™	ND	4T1	1 MHz, 2 W cm ⁻²	27
SDT + PTT	TiO ₂ + graphene	4T1	4T1	1 MHz, 1.5 W cm ⁻²	29	
	Black TiO _{2-x}	4T1	4T1	1 MHz, 1.5 W cm ⁻²	30	
	TiO ₂ + Au	HeLa	HeLa	3 MHz, 1.5 W cm ⁻²	31	
	TAPP + Pt-CuS	CT26	CT26	1 MHz, 0.5 W cm ⁻²	32	
SDT + Chemo	Ce6 + TPZ	B16F10	B16F10	3 MHz, 1 W cm ⁻²	34	
	HMTNP + HCQ	MCF-7/MDA-MB-231/Hs538Bst/ HepG2/NIH3T3/Raw264.7	MCF-7/HepG2	1 W cm ⁻²	35	
	DSTN + Curcumin	HeLa/A549	ND	1 MHz, 2 W cm ⁻²	36	
	TiO ₂ + DOX	KB/MCF-7	S180	1 W cm ⁻²	37	
	Ce6 + DTX	B16F10	B16F10	1 MHz, 1 W cm ⁻²	38	
	ICG + DOX	4T1	4T1	1 MHz, 2-6 W cm ⁻²	39	
	PPIX + DOX	HCC/SMMC-7721	HCC	1 MHz, 1.5 W cm ⁻²	40	
	PGL + CPT + FUDR	HT-29	HT-29	1 MHz, 1 W cm ⁻²	41	
	TPPS + DOX + Catalase	HUVEC/MCF-7	MCF-7	1 MHz, 1 W cm ⁻²	42	
	TiO ₂ + PPY + HNK	4T1/HepG2	4T1	1 MHz, 1.5 W cm ⁻²	43	
SDT + CDT/gas therapy	PtCu ₃	4T1/HUVECs	4T1	35 kHz, 3 W cm ⁻²	44	
	HMME + MCC	MCF-7/NIH3T3	MCF-7	1 MHz, 1 W cm ⁻²	45	
	AIPH	MCF-7	MCF-7	1 MHz, 2.5 W cm ⁻²	46	
SDT + immuno therapy	HMTNP + TPZ + R-SNO	MCF-7	MCF-7	1 MHz, 1 W cm ⁻²	47	
	aPD-1	CT26	CT26	1 MHz	48	
	HMME + R837	4T1	4T1	1 MHz, 1.5 W cm ⁻²	49	
	TPPS + MEDI4893	MRSA	MRSA	1 MHz, 0.97 W cm ⁻²	50	

Copy from the Ref [1]

[1] Son, S.; Kim, J. H.; Wang, X.; Zhang, C.; Yoon, S. A.; Shin, J.; Sharma, A.; Lee, M. H.; Cheng, L.; Jiasheng Wu, J.; Kim, J. S., Multifunctional sonosensitizers in sonodynamic cancer therapy. *Chem. Soc. Rev.*, **2020**, 49, 3244-3261.

Reviewers' Comments:

Reviewer #3:

Remarks to the Author:

This study pioneered the use of $[\text{Ru}(\text{bpy})_3]^{2+}$ as a kind of sonosensitizer and tried to demonstrate that $[\text{Ru}(\text{bpy})_3]^{2+}$ -mediated sonodynamic therapy effectively killed the breast cancer cells in vitro and in vivo. The author has tried to optimize the ultrasound parameters for cell and animal experiments. However, from the perspective of unity and rigor of the study, the manuscript still need to be improved by considering the following revisions.

1. The results in Figure 1 have not yet involved cell experiments and studies of NADH, it might be more appropriate to delete the cells in Figure 1d and move Figure 1d to Figure 2.
2. Considering that group "Ru+NaN₃+US" has been added to Figure 3c and Figure 3d, and the whole experiment has been repeated, the old pictures of the other four groups in Figure 3c and Figure 3d should be replaced with the new ones.
3. It's difficult to say whether the pork used in Figure 4b belonged to necrotic tissue in which most cells have been died? It might be more appropriate to use other materials instead of pork to carry out this experiment.
4. I would like to know about the ultrasonic duty cycle used for cell and animal experiments in this study?
5. Given that the time interval between $[\text{Ru}(\text{bpy})_3]^{2+}$ injection and ultrasound treatment of the mice has been changed from 10 minutes to 4 hours, the images used in Figure 6b, Supplementary Figure 14 and Supplementary Figure 15 should be updated.
6. The lungs do not seem to be the only common site of breast cancer metastasiss. Is it possible to observe a decrease of lung metastasis accompanied by an increase of other organ matastasis?
7. It is obvious that Figure 2a, Supplementary Figure 7 and Supplementary Figure 8 discuss the same problem. I wonder the basis for this sort of ordering.
8. It might be better to adjust the order of Supplementary Figure 13 immediately after Supplementary Figure 8.
9. The Supplementary Figure 9a, Supplementary Figure 9b, Supplementary Figure 10c and Supplementary Figure 13 needs to be marked with statistical significance.
10. " $[\text{Ru}(\text{bpy})_3]^{2+}$ " in the figures were labeled as "Ru", this marking method is worth discussing.
11. If possible, please add some studies to exclude the effects of SDT on other kinds of ROS.

Thank you very much for your email of our third revision. We are very sorry that we took a bit long time for each revision and supplement of the experiments because of the epidemic.

We are deeply thankful for the Reviewer 3's insightful comments and suggestions. All questions raised by Reviewer 3 are answered sincerely. The following are our point-to-point responses to the concerns and revisions made accordingly in the text and SI.

Reply to Reviewer 3:

This study pioneered the use of $[Ru(bpy)_3]^{2+}$ as a kind of sonosensitizer and tried to demonstrate that $[Ru(bpy)_3]^{2+}$ -mediated sonodynamic therapy effectively killed the breast cancer cells in vitro and in vivo. The author has tried to optimize the ultrasound parameters for cell and animal experiments. However, from the perspective of unity and rigor of the study, the manuscript still need to be improved by considering the following revisions.

Reply to the reviewer: We thank the reviewer for these meticulous comments. We revise our manuscript and add more experiments to improve the unity and rigor of this study.

1). *The results in Figure 1 have not yet involved cell experiments and studies of NADH, it might be more appropriate to delete the cells in Figure 1d and move Figure 1d to Figure 2.*

Reply to the reviewer: We thank the reviewer for this suggestion. We have updated Figure 1 and Figure 2.

Figure 1

Figure 2.

2). Considering that group “Ru+NaN₃+US” has been added to Figure 3c and Figure 3d, and the whole experiment has been repeated, the old pictures of the other four groups in Figure 3c and Figure 3d should be replaced with the new ones.

Reply to the reviewer: We apologize for our mistakes and thank the reviewer for this suggestion. For Figure 3c, we replaced with the new ones in the whole experiments in this revision.

For Figure 3d, it originally contained “Ru+NaN₃+US” group in our first manuscript. We did not repeat in the second revision.

Figure 3c

3). *It's difficult to say whether the pork used in Figure 4b belonged to necrotic tissue in which most cells have been died? It might be more appropriate to use other materials instead of pork to carry out this experiment.*

Reply to the reviewer: We thank the reviewer for raising this important suggestions. In the light of our present knowledge, the strength of structure of pork is similar to muscle tissue of human. The pork is the most suitable materials which we can obtain reasonably. This method is also reported by many references (Refs: Adv. Mater. 2015, 27, 6820–6827. J Med Ultrasonics 2009, 36, 53–60. Phys. Med. Biol. 2000, 45, 1511). If we use live tissue with 10 cm depth, only the large animals (such as monkeys) can meet the requirement. It is difficult to us at the moment. We apologize for this.

4). *I would like to know about the ultrasonic duty cycle used for cell and animal experiments in this study? .*

Reply to the reviewer: The ultrasound used in this article is all 10% duty cycle. This explanation has added in the “Instruments” part in our supporting information. “The ultrasound used in this article is 10% duty cycle, and the marked power is the average output power, which is calculated by multiplying the working power by 10%.”

5). *Given that the time interval between [Ru(bpy)₃]²⁺ injection and ultrasound treatment of the mice has been changed from 10 minutes to 4 hours, the images used in Figure 6b, Supplementary Figure 14 and Supplementary Figure 15 should be updated.*

Reply to the reviewer: We thank the reviewer for these critiques. Follow your suggestions, we have updated Figure 6b and Supplementary Figure 14. In Figure 6b, we horizontally sliced the collected lung to reveal deep lung tissue using a tissue slicer.

Supplementary Figure 15 is the biosafety and the acute toxicity of [Ru(bpy)₃]²⁺. In this experiment, [Ru(bpy)₃]²⁺ was i.v. injected into healthy mice. It is different experiments and conditions from Figure 6 and S14.

Figure 6

Figure S14

6). The lungs do not seem to be the only common site of breast cancer metastasis. Is it possible to observe a decrease of lung metastasis accompanied by an increase of other organ metastasis?

Reply to the reviewer: Thank you for this suggestion. As we know, after entering into the blood circulation system from tumor capillaries, circulating tumor cells (CTCs) first get to the right heart along the direction of the venous blood flow. Then, CTCs flows to the lung tissue with abundant capillaries. Narrow and dense capillary bed of alveoli capture CTCs. Thus, generally speaking, lung

tissue is the most serious site of breast cancer metastasis. Our first author Liang Chao's previous review described this process in detail (Chem. Soc. Rev., 2016, 45, 6250-6269). In our opinion, it is impossible to observe a decrease of lung metastasis. But, it is possible to observe an increase of other organ metastasis. During our experiment, we tried to find visible metastases in other organ (including liver, brain and bone). The result was that only slight metastases of liver was found in few mouse (please see the following picture).

7). *It is obvious that Figure 2a, Supplementary Figure 7 and Supplementary Figure 8 discuss the same problem. I wonder the basis for this sort of ordering.*

Reply to the reviewer: Supplementary Figure 7 and Supplementary Figure 8 are the control groups of Figure 2a. Supplementary Figure 7 is the control groups which lacked some necessary experimental conditions compared to Figure 2a. Supplementary Figure 7 and Figure 2a proved that NADH oxidation need both $[\text{Ru}(\text{bpy})_3]^{2+}$ and US irradiation. Figure 2a and Supplementary Figure 8 proved that NADH oxidation and singlet oxygen production are non-interfering. The methods of experiment design refer to our previous work (Nature Chemistry 2019, 11, 1041–1048).

8). *It might be better to adjust the order of Supplementary Figure 13 immediately after Supplementary Figure 8.*

Reply to the reviewer: Thank you very much for this suggestion. Follow your suggestion, we have adjusted the Figure S13 after Figure S8.

9). *The Supplementary Figure 9a, Supplementary Figure 9b, Supplementary Figure 10c and Supplementary Figure 13 needs to be marked with statistical significance.*

Reply to the reviewer: We apologize for our error. We have added the statistical significance in these Figures.

10). *“[Ru(byp)3]2+” in the figures were labeled as “Ru”, this marking method is worth discussing.*

Reply to the reviewer: In order to reduce the space in the Figures, we used the abbreviation “**Ru**” meant “[Ru(bpy)₃]²⁺”, we have annotated this in the figure caption and text: “The **Ru** is [Ru(bpy)₃]²⁺”. We hope you can understand this abbreviation.

11). *If possible, please add some studies to exclude the effects of SDT on other kinds of ROS.*

Reply to the reviewer: We thank the reviewer for this suggestion. As shown in Figure 1a, no obvious DMPO signal was observed. Superoxide radical (O₂^{•-}) and hydroxyl radical (HO•) could be captured by DMPO, thus this experiment could exclude the effects of both O₂^{•-} and HO•. In addition, in our first responding letter, as the reviewer’s suggestion, we had detected O₂^{•-} by using DHE probe and HO• by using HPF probe, respectively. The results showed that O₂^{•-} and HO• did not be found in the cells treated with [Ru(bpy)₃]²⁺ and US irradiation.

Finally, we would like to thank the referee again for taking the time to review our manuscript.
We sincerely hope that this paper can be accepted after this revision.

Best wishes

Pingyu

Reviewers' Comments:

Reviewer #3:

Remarks to the Author:

This study pioneered the use of $[\text{Ru}(\text{bpy})_3]^{2+}$ as a kind of sonosensitizer and tried to demonstrate that $[\text{Ru}(\text{bpy})_3]^{2+}$ -mediated sonodynamic therapy effectively killed the breast cancer cells in vitro and in vivo. The author has tried to optimize the ultrasound parameters for cell and animal experiments. However, a few minor issues still need to be addressed in order for this study to be accepted.

1. As for the experiment to demonstrate the strong penetration of ultrasound in Figure 4b, non-biological materials such as resins and gels are also worth considering in the future research.
2. The Supplementary Figure 10a still need to be marked with statistical significance.
3. The results may be more complete if the figures in "reply to question 11" can be added to the supplementary figures.

Reply to Reviewer 3:

This study pioneered the use of $[Ru(bpy)_3]^{2+}$ as a kind of sonosensitizer and tried to demonstrate that $[Ru(bpy)_3]^{2+}$ -mediated sonodynamic therapy effectively killed the breast cancer cells in vitro and in vivo. The author has tried to optimize the ultrasound parameters for cell and animal experiments. However, a few minor issues still need to be addressed in order for this study to be accepted.

Reply to the reviewer: Thank you for the enthusiastic comments. We revise our manuscript as your following suggestions.

1). *As for the experiment to demonstrate the strong penetration of ultrasound in Figure 4b, non-biological materials such as resins and gels are also worth considering in the future research.*

Reply to the reviewer: We thank you for this suggestion. In the future research, we would consider introducing mimic tissue made of resins and gels.

2). *The Supplementary Figure 10a still need to be marked with statistical significance.*

Reply to the reviewer: Thank you for this suggestion. We have added the statistical significance in Supplementary Figure 10a.

3). *The results may be more complete if the figures in “reply to question 11” can be added to the supplementary figures.*

Reply to the reviewer: Thank you for this suggestion. We have added Supplementary Figure 14 in the SI. And we have updated the following discussion in the text. “To investigate the kinds of intracellular ROS of $[Ru(bpy)_3]^{2+}$ for sonotherapy, dihydroethidium (DHE) and 3’-

hydroxy-6'-(4-hydroxyphenoxy)spiro[2-benzofuran-3,9'-xanthene]-1-one (HPF) staining assays were used to capture superoxide anion ($O_2^{\cdot-}$) and hydroxyl radical ($\cdot OH$), respectively (Supplementary Fig. 14). No obvious DHE signal or HPF signal could be found in the 4T1 cells treated with $[Ru(bpy)_3]^{2+}$ and US irradiation. These results excluded the effects of SDT on $O_2^{\cdot-}$ and $\cdot OH$. ”

Supplementary Figure 14. Confocal images of 4T1 cells stained with DHE or HPF after various treatments. The experiment was repeated three times independently with similar results. Ru: $10 \mu M [Ru(bpy)_3]^{2+}$; US: $0.3 W cm^{-2}$, 3 MHz, 20 min.